# FORCE-GUIDED BRIDGE MATCHING FOR FULL-ATOM TIME-COARSENED MOLECULAR DYNAMICS

## ABSTRACT

Molecular Dynamics (MD) is crucial in various fields such as materials science, chemistry, and pharmacology to name a few. Conventional MD software struggles with the balance between time cost and prediction accuracy, which restricts its wider application. Recently, data-driven approaches based on deep generative models have been devised for time-coarsened dynamics, which aim at learning dynamics of diverse molecular systems over a long timestep, enjoying both universality and efficiency. Nevertheless, most current methods are designed solely to learn from the data distribution regardless of the underlying Boltzmann distribution, and the physics priors such as energies and forces are constantly overlooked. In this work, we propose a conditional generative model called Force-guided Bridge Matching (FBM), which learns full-atom time-coarsened dynamics and targets the Boltzmann-constrained distribution. With the guidance of our delicately-designed intermediate force field, FBM leverages favourable physics priors into the generation process, giving rise to enhanced simulations. Experiments on two datasets consisting of peptides verify our superiority in terms of comprehensive metrics and demonstrate transferability to unseen systems.

## 1 INTRODUCTION

Molecular Dynamics (MD), which simulates the physical movements of molecular systems at the atomic level via *in silico* methods, are widely applied in the fields of materials science, physics, chemistry, and pharmacology (Wolf et al., 2005; Durrant & McCammon, 2011; Salo-Ahen et al., 2020). Accurate MD simulations enable the researcher to comprehend the equilibrium thermodynamics and kinetics of different molecular phases without the need for expensive wet-lab experiments.

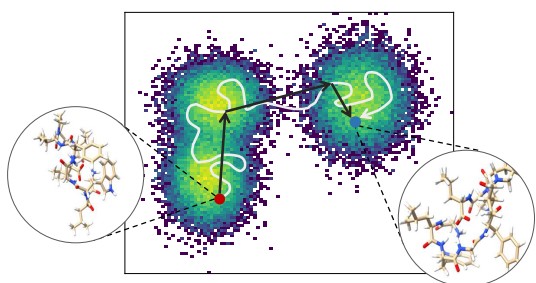

Figure 1: Illustration of how molecular conformations transfer from one state to another by MD (path in white) and time-coarsened dynamics (path in black).

Conventional MD software, like AMBER (Pearlman et al., 1995) and CHARMM (Vanommeslaeghe et al., 2010), mostly pre-defines an empirical force field of molecular systems and performs MD based on the numerical integration of Newtonian equations over the timestep $\Delta t$. To minimize discretization errors, $\Delta t$ should be chosen small enough, typically on the order of femtoseconds ($10^{-15}$s). Consequently, simulating the full duration of critical phase transitions that occur on the microsecond ($10^{-6}$s) or even millisecond ($10^{-3}$s) timescale becomes virtually impossible within a reasonable wall-clock time. To overcome the efficiency limitation in long-timescale simulation, a surge of approaches have been devised, including MD-like methods (Voter et al., 2000; Pang et al., 2017; Laio & Parrinello, 2002) and Monte Carlo-based methods (Sadigh et al., 2012; Sugita & Okamoto, 1999; Neyts & Bogaerts, 2014). However, all these methods share a common drawback: different molecular systems require customized simulations, despite the fact that many atomistic systems should, in principle, exhibit similar dynamic mechanisms.

Recently, data-driven approaches (Klein et al., 2024; Schreiner et al., 2024; Hsu et al., 2024; Li et al., 2024; Jing et al., 2024) leveraging deep generative models have been used to enhance MD simulations. Unlike traditional numerical integration, these methods directly learn MD from diverse observed trajectories, offering superior transferability across different molecular systems. Additionally, they can accelerate simulations through the use of *time-coarsened dynamics*, where models learn to generate new states after a significantly larger timestep ($\tau \gg \Delta t$) starting from an initial state, as illustrated in Figure 1. Nevertheless, existing learning-based methods still face two significant issues. The first issue is that most current methods are designed solely to learn from the data distribution, which may be biased compared to the underlying Boltzmann distribution (Boltzmann, 1877) — a fundamental concept for describing the thermal equilibrium of molecular systems in relation to atomic positions and velocities. Although Timewarp (Klein et al., 2024) alleviates this issues by employing the Metropolis-Hastings algorithm (Metropolis et al., 1953) to resample from the equilibrium, the unbearably low acceptance rate still hinders its applicability. The second issue is that physics priors (*e.g.*, energies and forces) are constantly overlooked during the learning process, which, yet, are critical for providing insights into the dynamics of molecular systems.

In this work, we delicately design a generative model to learn time-coarsened dynamics, and target the Boltzmann-constrained distribution $p(\vec{\mathbf{X}})$. This distribution incorporates a potential energy term $\exp(-k\varepsilon(\vec{\mathbf{X}}))$, into the original data distribution $q(\vec{\mathbf{X}})$, resulting in $p(\vec{\mathbf{X}}) = \frac{1}{Z}q(\vec{\mathbf{X}})\exp(-k\varepsilon(\vec{\mathbf{X}}))$. Here, $\vec{\mathbf{X}}$ stands for the molecular conformation, and $Z$ is the normalization factor. The definition of $p(\vec{\mathbf{X}})$ ensures that regions of high density in the generated distribution correspond to low potentials $\varepsilon(\vec{\mathbf{X}})$, thereby aligning more closely with thermodynamic principles. To learn $p(\vec{\mathbf{X}})$, we utilize the *bridge matching* generative framework (Shi et al., 2024) in order to make the generation to be conditioned on the starting molecular conformations rather than Gaussian priors. The most challenging part is that, the vector field to generate $p(\vec{\mathbf{X}})$ is strongly associated with the virtual "energies" and "forces" during the generation process, which, yet, are nontrivial to obtain. To address the issue, we derive an effective and rigorous way to interpolate a well-designed *intermediate force field* into the bridge matching framework based on reasonable assumptions. The proposed framework is termed as Force-guided Bridge Matching (FBM). It is also worth noting that FBM employs `TorchMD-NET` (Pelaez et al., 2024) as the backbone model, which enables full-atom modeling while preserving $\mathrm{SO}(3)$-equivariance. Our main contributions are summarized as follows:

1. To our best knowledge, FBM is the first full-atom generative model that directly targets the Boltzmann-constrained distribution without extra resampling steps, which is tailored for time-coarsened dynamics.

2. Under rigorous theoretical derivation, we integrate a well-defined intermediate force field into the bridge matching framework, which effectively involves the guidance of physics priors for better MD simulations.

3. We evaluate FBM on two datasets consisting of small peptides, where FBM exhibits transferability to unseen peptide systems and consistently showcases state-of-the-art results across various peptides.

## 2 RELATED WORK

**Boltzmann Generator**  An important objective of MD research is to quickly sample from the Boltzmann distribution, thereby revealing the free energy landscape and collective variables of matter of molecular systems. *Boltzmann generators* (Noé et al., 2019; Köhler et al., 2021; 2023; Falkner et al., 2023; Klein et al., 2024; Klein & Noé, 2024) employ generative models to produce samples asymptotically from the Boltzmann distribution mainly by: (i) Apply reweighting techniques to i.i.d. generated samples. (ii) Inference in a Markov Chain Monte Carlo (MCMC) procedure. These approaches heavily rely on MC resampling techniques, which are the bottleneck of the sampling efficiency due to costly energy calculation and low acceptance rates. Most similar to our work, Wang et al. (2024) propose ConfDiff that incorporate the energy and force guidance directly into score diffusion to target the Boltzmann-constrained distribution, yet it only works well on protein backbones and generate conformations in an unconditional way that fails to capture temporal dynamics. In contrast, FBM introduces the force guidance into the bridge matching framework, making it possible to sample straightforwardly from the Boltzmann-constrained distribution.

**Time-Coarsened Dynamics**  To overcome the instability of numerical integration of conventional MD simulations, many deep learning methods have adopted the fashion of *time-coarsened dynamics*, where models learn dynamics of diverse molecular systems over a long timestep. Fu et al. (2023) proposes a multi-scale graph network to learn dynamics of polymers, which fails to operate in the all atom system. ITO (Schreiner et al., 2024) is devised to learn the transition probability over multiple time resolutions, yet its transferability across chemical space remains unknown. Recently, Timewarp (Klein et al., 2024) and TBG (Klein & Noé, 2024) utilize augmented normalizing flows and flow matching respectively for transferable time-coarsened dynamics of small peptides, while both need additional reweighting procedures to debias expectations to the Boltzmann distribution. In addition, Score Dynamics (Hsu et al., 2024) applies score diffusion to capture dynamic patterns merely from the data distribution, while F$^3$low (Li et al., 2024) and MDGen (Jing et al., 2024) learn transformations of proteins under coarse-grained representations. On the contrary, FBM is designed to learn time-coarsened dynamics at the atomic level and target the Boltzmann-constrained distribution by directly inference without extra steps, which aligns well to thermodynamic principles and exhibits transferability to unseen peptide systems as well.

## 3  METHOD

In this section, we will present the overall workflow of our method. Specifically, in § 3.1, we will define our task formulation of time-coarsened dynamics, with providing necessary notations. Then in § 3.2, we introduce how to learn the time-coarsened dynamics from the data distribution, via a conditional generative model based on bridge matching, which is dubbed as FBM-BASE. Built upon this baseline, in § 3.3, we further propose a novel generative framework that targets the Boltzmann-constrained distribution by incorporating an intermediate force field into the FBM-BASE model, leading to our eventual model FBM. All proofs of propositions are provided in § B.

### 3.1  MOLECULAR REPRESENTATION

We represent each molecule (namely peptide in our experiments) as a graph $\mathcal{G} = (\mathcal{V}, \mathcal{E})$ consisting of the node set $\mathcal{V}$ and the edge set $\mathcal{E}$. For a molecule with $N$ atoms (including hydrogens), $\mathcal{V} = \{v_0, \cdots, v_{N-1}\}$ where $v_i$ $(0 \leq i < N)$ represents the $i$-th atom of the molecule. Each node $v_i$ is further attributed with Cartesian coordinates $\vec{x}_i \in \mathbb{R}^3$ from the structural information and node features $z_i \in \mathbb{R}^H$ from the embedding of atom types, where $H$ represents the hidden dimension. Particularly for peptides that are composed of 20 natural amino acids and exhibit similar features, we construct the atom type vocabulary based on the atom nomenclature of Protein Data Bank (Berman et al., 2000) to obtain more refined atomic representations. Formally, the features and Cartesian coordinates of all nodes are concatenated as node representations:

$$\vec{X} = [\vec{x}_0, \cdots, \vec{x}_{N-1}]^\top \in \mathbb{R}^{N \times 3}, \ Z = [z_0, \cdots, z_{N-1}]^\top \in \mathbb{R}^{N \times H}. \tag{1}$$

The edges are constructed with the cutoff radius $r_{\text{cut}}$. For any node pair $(v_i, v_j)$, the connection is established iff $||\vec{x}_i - \vec{x}_j|| < r_{\text{cut}}$. Constructing cutoff graphs is a favorable choice for modelling molecular systems since forces and chemical bonds are highly related to interatomic distances. With the aforementioned notations, we provide the task formulation below.

**Task Formulation**  Given each MD trajectory, we extract molecule pairs $\mathcal{D} = \{(\mathcal{G}_s, \mathcal{G}_{s+\tau}) \mid s \in \mathcal{S}\}$ to create a training dataset, where $\mathcal{G}_s$ denotes the starting state at time $s$ and $\mathcal{G}_{s+\tau}$ represents the future state after a temporal interval $\tau$. Our goals are: (i) Train a *baseline* model that fits the conditional data distribution $\mu(\vec{X}_{s+\tau}|\vec{X}_s)$ from $\mathcal{D}$, denoted as $q$. (ii) Based on the trained baseline model, we further train a new model that admits the Boltzmann constraint: $p(\vec{X}) = \frac{1}{Z}q(\vec{X})\exp(-k\varepsilon(\vec{X}))$, where $k$, $\varepsilon$, and $Z$ denote the inverse temperature, the potential of the molecular system, and the partition function, respectively. During the training process, we require the queries of the MD energies and forces for each pair $(\mathcal{G}_s, \mathcal{G}_{s+\tau})$, namely, $(\varepsilon(\vec{X}_s), \varepsilon(\vec{X}_{s+\tau}))$ and $(\nabla\varepsilon(\vec{X}_s), \nabla\varepsilon(\vec{X}_{s+\tau}))$. In the following context, we will train three neural networks $v_\theta$, $u_\theta$, and $w_\theta$ to approximate the vector fields that are used to generate the distributions $q$ and $p$.

## 3.2 BASELINE BRIDGE MATCHING

We will present the baseline BM model (*i.e.*, FBM-BASE) to fit the conditional data distribution $\mu(\vec{X}_{s+\tau}|\vec{X}_s)$ from dataset $\mathcal{D}$. For the sake of convenience and consistency with the continuous diffusion time between $[0, 1]$, we refer to the prior graph as $\mathcal{G}_0$ and the target graph as $\mathcal{G}_1$ in what follows, and denote their corresponding coordinates as $\vec{X}_0$ and $\vec{X}_1$ accordingly.

**Bridge Matching Framework**   We leverage the generative framework of *bridge matching* (Shi et al., 2024), which learns a mimicking diffusion process between two arbitrary distributions, allowing for more flexible choices of priors. Let $q_0, q_1$ denote the prior and target data distributions, respectively, and $t \in [0, 1]$ be the continuous diffusion time. We define the forward process:

$$\mathrm{d}\vec{\mathbf{X}}_t = f_t(\vec{\mathbf{X}}_t)\,\mathrm{d}t + \sigma_t\,\mathrm{d}\mathbf{B}_t,\ \vec{\mathbf{X}}_0 \sim q_0, \tag{2}$$

where $\vec{\mathbf{X}}_t$ and $\mathbf{B}_t$ represent the random variable and the Brownian motion of the diffusion process at time $t$, respectively. With the process pinned down at an initial point $\vec{X}_0$ and terminal point $\vec{X}_1$, the conditional distribution $q_t(\cdot|\vec{X}_0, \vec{X}_1)$ will be realized as a *diffusion bridge* in the form of $\mathrm{d}\vec{\mathbf{X}}_t = \{f_t(\vec{\mathbf{X}}_t) + \sigma_t^2 \nabla \log q_t(\vec{X}_1|\vec{\mathbf{X}}_t)\}\,\mathrm{d}t + \sigma_t\,\mathrm{d}\mathbf{B}_t$ with $\vec{\mathbf{X}}_0 = \vec{X}_0$, where Doob $h$-transform theory (Rogers & Williams, 2000) guarantees $\vec{\mathbf{X}}_1 = \vec{X}_1$. For simplification, by considering $f_t = 0$ and $\sigma_t = \sigma$, the process will degenerate to the following *Brownian bridge*:

$$\mathrm{d}\vec{\mathbf{X}}_t = \frac{\vec{X}_1 - \vec{\mathbf{X}}_t}{1 - t}\,\mathrm{d}t + \sigma \mathrm{d}\mathbf{B}_t,\ \vec{\mathbf{X}}_0 = \vec{X}_0, \tag{3}$$

which yields the conditional distribution $q_t(\vec{\mathbf{X}}_t|\vec{X}_0, \vec{X}_1)$ at time $t \in [0, 1]$, with its marginal defined as $q_t(\vec{\mathbf{X}}_t)$. The core of bridge matching is to find a Markov diffusion governed by a *vector field* $v$:

$$\mathrm{d}\vec{\mathbf{X}}_t = v(\vec{\mathbf{X}}_t, t)\,\mathrm{d}t + \sigma \mathrm{d}\mathbf{B}_t, \tag{4}$$

which admits the same marginal $\vec{\mathbf{X}}_t \sim q_t$, such that $\vec{\mathbf{X}}_1 \sim q_1$ holds. To achieve this, we can learn a parametric vector field $v_\theta$ via the following regression loss:

$$\mathcal{L}_{\mathrm{fwd}} = \mathbb{E}_{t \sim \mathrm{Uni}(0,1),(\mathcal{G}_0,\mathcal{G}_1)\sim\mathcal{D},\vec{X}_t \sim q_t(\cdot|\vec{X}_0,\vec{X}_1)}[||\frac{\vec{X}_1 - \vec{\mathbf{X}}_t}{1 - t} - v_\theta(\vec{\mathbf{X}}_t, t)||^2], \tag{5}$$

where $\mathrm{Uni}(0, 1)$ denotes the uniform distribution of $[0, 1]$, and $v_\theta$ is implemented by a neural network parameterized by $\theta$. The calculation of Eq. (5) requires sampling from $q_t(\vec{\mathbf{X}}_t|\vec{X}_0, \vec{X}_1)$, which usually needs additional SDE simulations. Fortunately, Eq. (3) enables a closed-form solution as follows:

$$q_t(\vec{\mathbf{X}}_t|\vec{X}_0, \vec{X}_1) = \mathcal{N}(t\vec{X}_1 + (1-t)\vec{X}_0, t(1-t)\sigma^2\boldsymbol{I}), \tag{6}$$

where we can sample $\vec{X}_t$ efficiently during each training step with any given $t$.

The optimal vector field $v_\theta$ to minimize the loss in Eq. (5) is actually equal to the expectation $v^*(\vec{\mathbf{X}}_t, t) = \mathbb{E}_{q_t(\cdot,\cdot|\vec{\mathbf{X}}_t)}[\frac{\vec{X}_1 - \vec{\mathbf{X}}_t}{1-t}]$. Then the distribution $q_t$ can be estimated by performing SDE sampling in Eq. (4), using $v_\theta$ as the vector field $v$. In our experiments, $v_\theta$ is built upon `TorchMD-NET` (Pelaez et al., 2024) for full-atom modeling while satisfying SO(3)-equivariance. Details of the architecture of neural networks used in our model are further elucidated in § C.

## 3.3 FORCE-GUIDED BRIDGE MATCHING

In most cases, the training datasets are biased from the underlying Boltzmann distribution, leading to defective prediction even for superior generative models. In this section, we will introduce how to train a force-guided generative model FBM, which admits the Boltzmann-constrained distribution denoted as $p_1(\vec{X}_1) = \frac{1}{Z}q_1(\vec{X}_1)\exp(-k\varepsilon(\vec{X}_1))$. Here, the exponential term $\exp(-k\varepsilon(\vec{\mathbf{X}}_1))$ serves as the physics prior from thermodynamic principles. The overall framework of FBM as well as FBM-BASE is illustrated in Figure 2.

Figure 2: The overall framework of FBM-BASE and FBM. **A.** Firstly, FBM-BASE leverages the bridge matching framework to learn time-coarsened dynamics from the data distributions $q_0$ and $q_1$. **B.** With the guidance of the intermediate force field $\nabla\varepsilon_t$ at diffusion time $t$, the marginal distribution admits $p_t(\vec{\mathbf{X}}_t) \propto q_t(\vec{\mathbf{X}}_t) \exp(-k\varepsilon_t(\vec{\mathbf{X}}_t))$, thereby the target distribution of FBM is debiased to the Boltzmann-constrained distribution $p_1$.

**Force-guided Bridge Matching Framework** In order to learn $p_1$ under the bridge matching framework, our key idea is to construct a new probabilistic path $p_t$ based on the existing probabilistic path $q_t$ from Eq. (3), such that the following condition is satisfied for $t \in [0,1]$:

$$p_t(\vec{\mathbf{X}}_t) = \frac{1}{Z_t} q_t(\vec{\mathbf{X}}_t) \exp(-k\varepsilon_t(\vec{\mathbf{X}}_t)), \ \varepsilon_0(\cdot) = \varepsilon_1(\cdot) = \varepsilon(\cdot). \tag{7}$$

Here $Z_t$ is the partition function and $\varepsilon_t$ is an artificially-designed *intermediate potential* of the process, which should converge to the real MD potential $\varepsilon$ when $t \to 0^+$ and $t \to 1^-$ for consistency. Further, we assume $p_t(\vec{\mathbf{X}}_t|\vec{X}_0, \vec{X}_1) = q_t(\vec{\mathbf{X}}_t|\vec{X}_0, \vec{X}_1)$ as in Eq. (6), thereby the stochastic process governed by $p_t$ shares the same form with Eq. (3), which can be modeled by the following Markov diffusion process associated with a vector field $v'$:

$$d\vec{\mathbf{X}}_t = v'(\vec{\mathbf{X}}_t, t) \, dt + \sigma d\mathbf{B}_t. \tag{8}$$

Therefore, we are able to inference $p_t$ if we know how to learn $v'$ from the dataset. Interestingly, we prove that $v'(\vec{\mathbf{X}}_t, t)$ can be expressed in terms of the vector field $v^*(\vec{\mathbf{X}}_t, t)$ generating $q_t$ and the intermediate force field $\nabla\varepsilon_t(\vec{\mathbf{X}}_t)$ allowing for the Boltzmann constraint, which will be formally demonstrated in Proposition 3.2. In prior to showing this proposition, we first derive the form of the intermediate force field below:

**Proposition 3.1.** *Assume that the joint distributions $q(\vec{\mathbf{X}}_0, \vec{\mathbf{X}}_1)$ and $p(\vec{\mathbf{X}}_0, \vec{\mathbf{X}}_1)$ satisfy $p(\vec{\mathbf{X}}_0, \vec{\mathbf{X}}_1) \propto q(\vec{\mathbf{X}}_0, \vec{\mathbf{X}}_1) \exp(-k(\varepsilon(\vec{\mathbf{X}}_0) + \varepsilon(\vec{\mathbf{X}}_1)))$, the intermediate force field $\nabla\varepsilon_t$ is given by:*

$$\nabla\varepsilon_t(\vec{\mathbf{X}}_t) = \frac{\mathbb{E}_{q(\vec{\mathbf{X}}_0, \vec{\mathbf{X}}_1)}[q_t(\vec{\mathbf{X}}_t|\vec{X}_0, \vec{X}_1) \exp(-k(\varepsilon(\vec{\mathbf{X}}_0) + \varepsilon(\vec{\mathbf{X}}_1)))\zeta(\vec{\mathbf{X}}_0, \vec{\mathbf{X}}_1, \vec{\mathbf{X}}_t)]}{k\mathbb{E}_{q(\vec{\mathbf{X}}_0, \vec{\mathbf{X}}_1)}[q_t(\vec{\mathbf{X}}_t|\vec{X}_0, \vec{X}_1) \exp(-k(\varepsilon(\vec{\mathbf{X}}_0) + \varepsilon(\vec{\mathbf{X}}_1)))]}, \tag{9}$$

*where we denote $\zeta(\vec{\mathbf{X}}_0, \vec{\mathbf{X}}_1, \vec{\mathbf{X}}_t) = \nabla \log q_t(\vec{\mathbf{X}}_t) - \nabla \log q_t(\vec{\mathbf{X}}_t|\vec{X}_0, \vec{X}_1)$ for brevity.*

**Proposition 3.2.** *Given the prerequisites in Proposition 3.1, we have $v'(\vec{\mathbf{X}}_t, t) = v^*(\vec{\mathbf{X}}_t, t) - \frac{\sigma^2}{2}k\nabla\varepsilon_t(\vec{\mathbf{X}}_t)$ under some mild assumptions.*

**Estimation of Immediate Force Field** However, it is challenging to calculate the intermediate force field $\nabla\varepsilon_t(\vec{\mathbf{X}}_t)$ owing to its nontrivial form in Eq. (9). Note that $q_t(\vec{\mathbf{X}}_t|\vec{X}_0, \vec{X}_1)$ can be directly calculated by Eq. (6), $\varepsilon(\vec{X}_0), \varepsilon(\vec{X}_1)$ are both known as MD potentials, and the the expectation in the denominator can be estimated with samples during a training mini-batch instead of the entire data distribution (Lu et al., 2023a). The most challenging part is the computation of the term $\nabla \log q_t(\vec{\mathbf{X}}_t)$ in $\zeta(\vec{\mathbf{X}}_0, \vec{\mathbf{X}}_1, \vec{\mathbf{X}}_t)$. To provide an unbiased estimation of this score, we investigate its relation with the vector fields. Given $\mathcal{G}_0, \mathcal{G}_1$, we first define the conditional score as $s_t(\vec{\mathbf{X}}_t|\vec{X}_0, \vec{X}_1) = \nabla \log q_t(\vec{\mathbf{X}}_t|\vec{X}_0, \vec{X}_1)$. Based on Eq. (6), the closed-form of $s_t$ is given by:

$$s_t(\vec{\mathbf{X}}_t|\vec{X}_0, \vec{X}_1) = -\frac{\vec{\mathbf{X}}_t - [t\vec{X}_1 + (1-t)\vec{X}_0]}{t(1-t)\sigma^2} = \frac{1}{\sigma^2}\left[\frac{\vec{X}_1 - \vec{\mathbf{X}}_t}{1-t} - \frac{\vec{\mathbf{X}}_t - \vec{X}_0}{t}\right]. \tag{10}$$

Note that the first term of Eq. (10) has the same form as in the training objective of Eq. (5). Similarly, we train another network $u_\theta$ to imitate the second term:

$$\mathcal{L}_{\text{rev}} = \mathbb{E}_{t \sim \text{Uni}(0,1), (\mathcal{G}_0, \mathcal{G}_1) \sim \mathcal{D}, \vec{\mathbf{X}}_t \sim q_t(\cdot | \vec{\mathbf{X}}_0, \vec{\mathbf{X}}_1)}[|| \frac{\vec{\mathbf{X}}_t - \vec{\mathbf{X}}_0}{t} - u_\theta(\vec{\mathbf{X}}_t, t) ||^2], \tag{11}$$

where the expectation of $u_\theta$ can be expressed as $u^*(\vec{\mathbf{X}}_t, t) = \mathbb{E}_{q_t(\cdot, \cdot | \vec{\mathbf{X}}_t)}[\frac{\vec{\mathbf{X}}_t - \vec{\mathbf{X}}_0}{t}]$. Then we take the expectation over $\vec{\mathbf{X}}_0, \vec{\mathbf{X}}_1$ conditioned on $\vec{\mathbf{X}}_t$ in Eq. (10), yielding:

$$s_t^*(\vec{\mathbf{X}}_t) = \mathbb{E}_{q_t(\cdot, \cdot | \vec{\mathbf{X}}_t)}[s_t(\vec{\mathbf{X}}_t | \vec{\mathbf{X}}_0, \vec{\mathbf{X}}_1)] = \frac{v^*(\vec{\mathbf{X}}_t, t) - u^*(\vec{\mathbf{X}}_t, t)}{\sigma^2}. \tag{12}$$

We present Proposition 3.3 to reveal that $s_t^*$ is identical to the marginal score $\nabla \log q_t$ of interest:

**Proposition 3.3.** *We have* $\nabla \log q_t(\vec{\mathbf{X}}_t) = s_t^*(\vec{\mathbf{X}}_t)$, *where* $\nabla \log q_t$ *is the score of the Brownian bridge defined in Eq. (3) and* $s_t^*(\vec{\mathbf{X}}_t)$ *is the expectation of the conditional score given by Eq. (12).*

In practice, $\nabla \log q_t(\vec{\mathbf{X}}_t)$ is estimated as $(v_\theta(\vec{\mathbf{X}}_t, t) - u_\theta(\vec{\mathbf{X}}_t, t))/\sigma^2$, by replacing the vector fields with the learned neural networks $v_\theta, u_\theta$ in Eq. (12). Now, since all quantities related to the intermediate force field $\nabla \varepsilon_t(\vec{\mathbf{X}}_t)$ are commutable, we then train a neural network $w_\theta(\vec{\mathbf{X}}_t, t)$ as its unbiased estimator:

$$\mathcal{L}_{\text{iff}} = \mathbb{E}_{t, (\mathcal{G}_0, \mathcal{G}_1), q_t(\cdot | \vec{\mathbf{X}}_0, \vec{\mathbf{X}}_1)}[|| \frac{\exp(-k(\varepsilon(\vec{\mathbf{X}}_0) + \varepsilon(\vec{\mathbf{X}}_1)))\zeta(\vec{\mathbf{X}}_0, \vec{\mathbf{X}}_1, \vec{\mathbf{X}}_t)}{k \mathbb{E}_{q(\vec{\mathbf{X}}_0, \vec{\mathbf{X}}_1)^B}[q_t(\vec{\mathbf{X}}_t | \vec{\mathbf{X}}_0, \vec{\mathbf{X}}_1) \exp(-k(\varepsilon(\vec{\mathbf{X}}_0) + \varepsilon(\vec{\mathbf{X}}_1)))]} - w_\theta(\vec{\mathbf{X}}_t, t) ||^2], \tag{13}$$

where $B$ denotes the mini-batch size of each training step.

## 3.4 Full Training Processes and Inference

In addition to the aforementioned regression losses, we introduce an auxiliary loss from Yim et al. (2023), which promotes predictions of pairwise atomic relations. The loss is defined as:

$$\mathcal{L}_{\text{aux}} = (1 - t) \cdot \frac{||\mathbf{1}_{\boldsymbol{D}_0 < 6\text{Å}}(\boldsymbol{D}_0 - \hat{\boldsymbol{D}}_0)||^2}{\sum \mathbf{1}_{\boldsymbol{D}_0 < 6\text{Å}} - N} + t \cdot \frac{||\mathbf{1}_{\boldsymbol{D}_1 < 6\text{Å}}(\boldsymbol{D}_1 - \hat{\boldsymbol{D}}_1)||^2}{\sum \mathbf{1}_{\boldsymbol{D}_1 < 6\text{Å}} - N}, \tag{14}$$

where $D_0, D_1 \in \mathbb{R}^{N \times N}$ denote pairwise distances between all atoms of $\mathcal{G}_0$ and $\mathcal{G}_1$, and $\hat{D}_0, \hat{D}_1$ are defined in the same way based on the estimated starting point $\hat{\vec{\mathbf{X}}}_0 = \vec{\mathbf{X}}_t - t u_\theta(\vec{\mathbf{X}}_t, t)$ and terminal point $\hat{\vec{\mathbf{X}}}_1 = \vec{\mathbf{X}}_t + (1 - t) v_\theta(\vec{\mathbf{X}}_t, t)$. The full loss of FBM-BASE is given by:

$$\mathcal{L}_{\text{base}} = \mathcal{L}_{\text{fwd}} + \mathcal{L}_{\text{rev}} + \lambda_{\text{aux}} \cdot \mathcal{L}_{\text{aux}}, \tag{15}$$

where $\lambda_{\text{aux}}$ is a hyper-parameter to balance the weight of different training objectives.

Empirically, large variances are noticed during training FBM with Eq. (13) when $t$ is close to 0 and 1. To address the issue, we find that the intermediate force field converges to the MD force field $\nabla \varepsilon$ at $t = 0, 1$, which is guaranteed by Proposition 3.4:

**Proposition 3.4.** *Given* $\varepsilon_0 = \varepsilon_1 = \varepsilon$ *and the intermediate force field described in Eq. (9), the continuity condition* $\lim_{t \to 0^+} \nabla \varepsilon_t(\vec{\mathbf{X}}_t) = \nabla \varepsilon(\vec{\mathbf{X}}_0)$, $\lim_{t \to 1^-} \nabla \varepsilon_t(\vec{\mathbf{X}}_t) = \nabla \varepsilon(\vec{\mathbf{X}}_1)$ *holds.*

Therefore, we leverage two separate networks $w_\theta^{(1)}, w_\theta^{(2)}$ to learn the boundary force fields:

$$\mathcal{L}_{\text{bnd}} = \mathbb{E}_{t, (\mathcal{G}_0, \mathcal{G}_1), q_t(\cdot | \vec{\mathbf{X}}_0, \vec{\mathbf{X}}_1)}[||\nabla \varepsilon(\vec{\mathbf{X}}_0) - w_\theta^{(1)}(\vec{\mathbf{X}}_t, t)||^2 + ||\nabla \varepsilon(\vec{\mathbf{X}}_1) - w_\theta^{(2)}(\vec{\mathbf{X}}_t, t)||^2]. \tag{16}$$

We construct the network $w_\theta$ in the interpolation form with another network $w_\theta^{(3)}$, similar to Máté & Fleuret (2023): $w_\theta(\vec{\mathbf{X}}_t, t) = (1 - t) w_\theta^{(1)}(\vec{\mathbf{X}}_t, t) + t w_\theta^{(2)}(\vec{\mathbf{X}}_t, t) + t(1 - t) w_\theta^{(3)}(\vec{\mathbf{X}}_t, t)$. The ultimate loss for training FBM is given by:

$$\mathcal{L}_{\text{FBM}} = \mathcal{L}_{\text{iff}} + \mathcal{L}_{\text{bnd}}. \tag{17}$$

**Full Training Processes** We first perform training to derive $v_\theta$ and $u_\theta$ under the base loss in Eq. (15), and then continue the training process to attain $w_\theta$ under the FBM loss in Eq. (17). The pseudo codes for training FBM-BASE and FBM are in Alg. 1 and Alg. 2, respectively.

**Force-guided Inference** For inference with FBM, we estimate the vector field $v'$ based on Proposition 3.2, by replacing $v^*$ and $\nabla \varepsilon_t$ with the trained neural networks $v_\theta$ and $w_\theta$. Force-guided inference is then performed following the SDE process of Eq. (8), where the diffusion time $t$ is discretized equidistantly. Additional details and pseudo codes for inference are provided in § D.

## 4 EXPERIMENTS

**Dataset Generation** We evaluate FBM on two datasets consisting of small peptides: *Alanine-Dipeptide (AD)* that is commonly studied previously and *PepMD* which is created by us. The AD dataset contains a simple peptide with only 22 atoms. The initial structure and reference MD trajectories of AD are all obtained from `mdshare`[1] without post-processing. As for PepMD, we first screen valid peptides between 3-10 residues from the sequence data provided by PDB (Berman et al., 2000). Next, we perform data cleaning according to the following criterion: each peptide must contain only the 20 natural amino acids, and the number of any type of residue should not exceed 50% of the sequence length. We then cluster the data with a sequence identity threshold of 60% by `MMseq2` (Steinegger & Söding, 2017), and randomly sample one peptide from each cluster to obtain a non-redundant dataset. Considering the computing resource constraints, we select 136/14 peptides for constructing the training-validation/test set, respectively. The structures of all 150 peptides are predicted by open-source tools `RDKit` (rdk) and `PDBfixer`[2], which are sent as initial states to generate MD trajectories using `OpenMM` (Eastman et al., 2017) afterwards. Finally, the peptide pairs for training are then sampled from trajectories in the way depicted in § 3.1. The MD simulation setups and the statistical details of our curated dataset are both illustrated in § E.1.

**Baselines** We compare our FBM with the following generative models that learns time-coarsened dynamics: (i) Timewarp (Klein et al., 2024), the current state-of-the-art model targeting the Boltzmann distribution by MCMC resampling, which exhibits superior transferability to unseen peptide systems. (ii) ITO (Schreiner et al., 2024), a conditional diffusion model that learns multiple time-resolution dynamics. (iii) Score Dynamics (Hsu et al., 2024), a score-based diffusion model that learns discrete transitions of the dynamic variables. All models are trained on PepMD from scratch for fair comparison.

**Metrics** Following Wang et al. (2024), we evaluate generated conformation ensembles against the full MD trajectories as to their validity, flexibility, and distributional similarity. We provide brief descriptions of the metrics in this part and further details are illustrated in § E.2:

- **Validity**. We regard a molecular conformation as *valid* when it is governed by certain physics constraints. Following Lu et al. (2023b), we judge whether a conformation is valid by the criterion: no bond clashes between any residue pairs and no bond breaks between adjacent residues, based on coordinates of $\alpha$-carbons. This metric, named as VAL-CA, represents the fraction of valid conformations in the full generated conformation ensembles.

- **Flexibility**. The generated structures are further required to exhibit flexibility to capture dynamic characteristics. Following Janson et al. (2023), we report the root mean square error of contact maps between generated conformation ensembles and reference MD trajectories as a measure of *flexibility*, termed as CONTACT.

- **Distributional similarity**. We focus on the similarity between the sample distribution and the Boltzmann distribution. Instead of the costly computation of the Boltzmann density, we project the molecular conformations onto following low-dimensional feature spaces and calculate the Jensen-Shannon (JS) distance as a substitute (Lu et al., 2023b): (i) pairwise distances between $\alpha$-carbons of residues (PWD); (ii) radius-of-gyration (RG) that measures the distribution of $\alpha$-carbons to the center-of-mass; (iii) the *time-lagged independent components* (Pérez-Hernández et al., 2013) (TIC) based on dihedrals and pairwise distances of $\alpha$-carbons, where only the slowest components, TIC 0 and TIC 1, are taken into consideration. For each metric, the mean JS distance along all feature dimensions are reported.

---

[1]https://github.com/markovmodel/mdshare
[2]https://github.com/openmm/pdbfixer

## 4.1 METASTABLE STATES EXPLORATION FOR AD

We first investigate how well the generated conformations can travel across different metastable states of AD. Due to the simple structure of AD with only one peptide bond, some metrics are not applicable except for TIC and TIC-2D (*i.e.*, the joint distribution of TIC 0 and TIC 1). In particular, the backbone dihedrals of AD, psi and phi, are commonly considered as two challenging variables for state transitions. Therefore, we include the similarity measurement of the joint distribution of psi and phi, *i.e.* the *Ramachandran plot* (Ramachandran et al., 1963), denoted as RAM.

In Table 1 we show evaluation results on AD, where models sample in the time-coarsened manner from the same initial state for a chain length of $10^3$. According to Table 1, FBM outperforms existing baselines on both RAM and TIC metrics, and with the introduction of physics priors, it shows considerable improvements in distribution similarity across various feature spaces compared to FBM-BASE. Although Timewarp surpasses FBM in the TIC-2D metric, we will explain later that it comes at the cost of generating invalid conformations.

Further, Ramachandran plots of generated ensembles are illustrated in Figure 3, where three known metastable states are recognized based on MD trajectories and labeled in order. Apparently, ITO and Score Dynamics fail to capture the dynamics of AD with samples randomly allocated. Moreover, Timewarp cannot rapidly traverse through metastable states, resulting in a great portion of invalid samples. Despite both FBM-BASE and FBM show relatively "clean" plots with fewer unreasonable conformations, FBM pays more attention to high density regions including cluster 2 and cluster 3, confirming a strong guidance of the intermediate force field to align more closely with thermodynamic principles.

Table 1: Results on alanine dipeptide. Values of each metric are averaged over three independent runs. The best result for each metric is shown in **bold** and the second best is underlined.

| MODELS | JS DISTANCE ($\downarrow$) | | |
|---|---|---|---|
| | RAM | TIC | TIC-2D |
| TIMEWARP | 0.722 | 0.546 | **0.719** |
| ITO | 0.740 | 0.696 | 0.833 |
| SD | 0.731 | 0.673 | 0.807 |
| FBM-BASE | 0.727 | 0.533 | 0.749 |
| FBM | **0.711** | **0.525** | 0.733 |

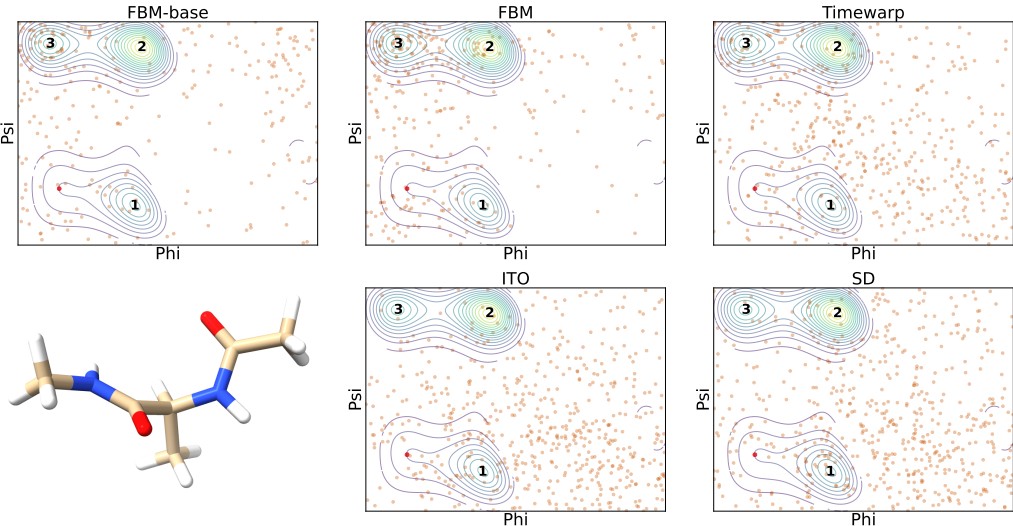

Figure 3: Ramachandran plots of alanine dipeptide with conformation ensembles generated by models. The initial state is indicated with the red cross. Contours represent the kernel densities estimated by the MD trajectory and the generated conformations are shown in scatter.

## 4.2 TRANSFERABILITY TO UNSEEN PEPTIDES OF PEPMD

We then explore the transferability of models to unseen peptides with various sequence lengths of PepMD. We use all metrics in § 4 for evaluation and the results on 14 test peptides of PepMD are demonstrated in Table 2, where all samples are generated for a chain length of $10^3$. We find Timewarp achieves a good performance on distributional similarity and flexibility, yet at the cost of only a small portion of valid samples. In contrast, our FBM showcases superiority across all metrics and achieves significant improvement on the validity of generated conformations in particular. It indicate that by introducing the force guidance, the generated ensembles of FBM better comply with the underlying Boltzmann distribution. Additional experimental results can be found in § F.

Table 2: Results on the test set of PepMD. Values of each metric are first averaged over 3 independent runs for each peptide and then shown in mean/std of all 14 test peptides. The best result for each metric is shown in **bold** and the second best is underlined.

| MODELS | JS DISTANCE (↓) | | | | VAL-CA (↑) | CONTACT (↓) |
| --- | --- | --- | --- | --- | --- | --- |
| | PWD | RG | TIC | TIC-2D | | |
| TIMEWARP | 0.575/0.082 | 0.561/0.124 | 0.633/0.069 | 0.804/0.025 | 0.115/0.121 | 0.197/0.128 |
| ITO | 0.833/0.000 | 0.829/0.012 | 0.789/0.067 | 0.833/0.000 | 0.001/0.000 | 0.940/0.081 |
| SD | 0.823/0.030 | 0.818/0.041 | 0.773/0.032 | 0.832/0.001 | 0.006/0.016 | 0.824/0.095 |
| FBM-BASE | 0.576/0.066 | 0.560/0.153 | 0.639/0.061 | 0.807/0.020 | 0.367/0.173 | 0.208/0.142 |
| FBM | **0.573**/0.064 | **0.542**/0.140 | **0.631**/0.077 | **0.801**/0.032 | **0.616**/0.188 | **0.188**/0.127 |

For better understanding, we provide the visualization of comprehensive metrics on the test peptide 1e28:C (TAFTIPSI) in Figure 4. In Figure 4(a), we show that the samples generated by FBM exhibit a more pronounced clustering in regions with high reference densities, though all compared methods inevitably generate samples in low-density regions. Figure 4(b) demonstrates that FBM accurately captures the peak of the distribution of the radius-of-gyration, with a discrepancy of less than 0.3Å in the right tail of the distribution. FBM and MD also show a close match in terms of the contact rate from Figure 4(c). Finally, according to Figure 4(d), we emphasize that FBM generates dominantly more valid conformations during the inference step compared to all baselines.

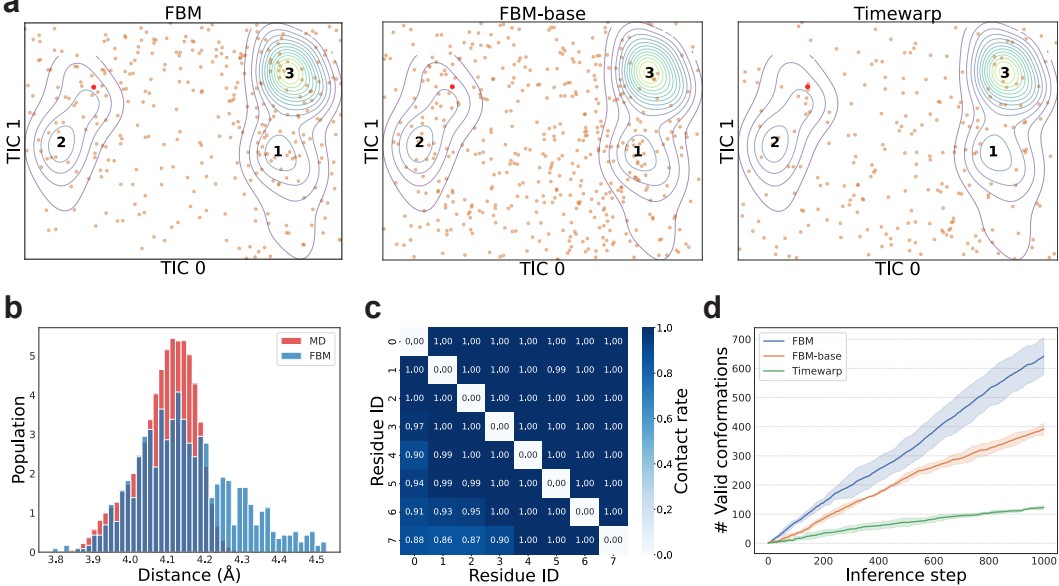

Figure 4: The visualization of comprehensive metrics on peptide 1e28:C. **a.** Plots of the slowest two TIC components analyzed by feature projections. **b.** The distribution of the radius-of-gyration. **c.** The residue contact map, where the data in the lower and upper triangle are obtained from FBM and MD, respectively. **d.** Cumulated valid conformations during inference over 3 independent runs.

In Figure 5, we conduct further comparisons with MD on conformation transitions over time. We first explore the ability of FBM to recover equilibrium conformations, which is measured by the lowest sample $C_\alpha$-RMSD to each cluster center (Wang et al., 2024). Reference structures and selected samples of FBM with the lowest $C_\alpha$-RMSD for 3 clusters of peptide 1e28:C are provided in Figure 5(a). The RMSD values of all pairs are below 2Å, showing a good recovery of representative conformations. In Figure 5(b), we provide the $C_\alpha$-RMSD values along trajectories compared with the initial state of peptide 1e28:C. Note that, since MD performs local energy minimization on the initial state before simulation, the starting point of its curve is not at the origin. We show that FBM gradually guides the peptide toward equilibrium, reaching a stable RMSD level similar to MD at around 70 ns. In contrast, FBM-BASE reaches a biased equilibrium at an early stage, while Timewarp exhibits excessive fluctuations over time. In Figure 5(c), we report the effective-sample-size per second of wall-clock time (ESS/s) (Klein et al., 2024) over the entire test set, where FBM achieves an efficiency improvement of around 10 times relative to MD based on the median values.

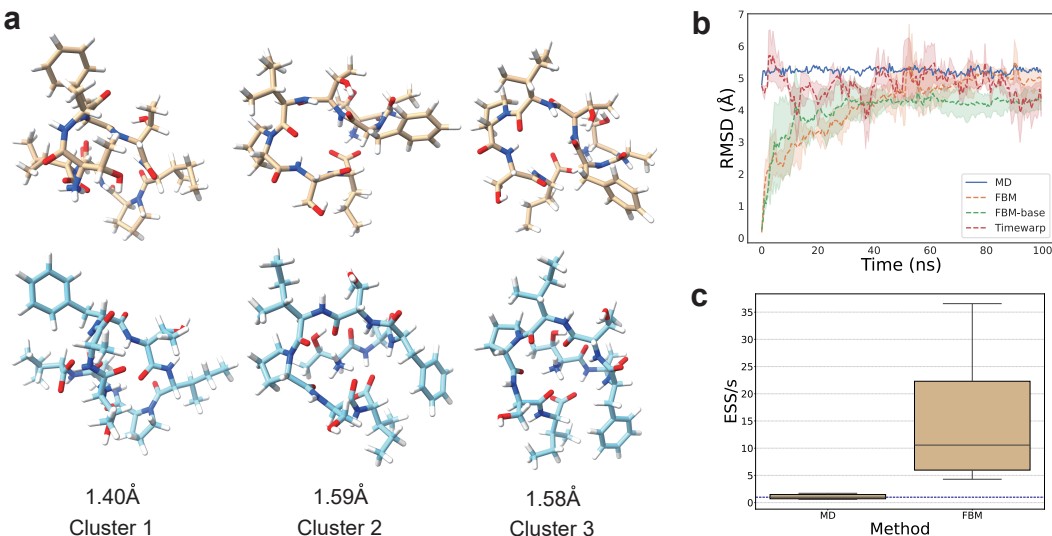

Figure 5: Comparisons between FBM and MD on conformation transitions over time. **a.** Comparison between the reference equilibrium conformations (blue) and the selected samples of FBM (yellow) of peptide 1e28:C. $C_\alpha$-RMSD values are reported below each cluster. **b.** $C_\alpha$-RMSD values along trajectories compared with the initial state of peptide 1e28:C over 3 independent runs. **c.** The effective sample size per second measured on the test set. All specific values are converted to multiples of the median value of MD, which is shown as the blue dashed line for reference.

## 5 CONCLUSION AND FUTURE WORK

In this work, we present a novel generative model called FBM for time-coarsened dynamics in a full-atom fashion. We first leverage the bridge matching framework to construct the baseline model FBM-BASE for learning dynamics from the data distribution. Based on FBM-BASE, we further introduce physics priors and interpolate a well-designed intermediate force field, which is theoretically guaranteed to target the Boltzmann-constrained distribution via directly inference without extra steps. Experiments on alanine dipeptide and our curated dataset PepMD showcase superiority of FBM on comprehensive metrics and demonstrate transferability to unseen peptide systems.

As the first attempt to incorporate the intermediate force field to bridge matching for full-atom time-coarsened dynamics, our method has considerable room for improvement. Firstly, our experiments have been conducted on small peptides with fewer than 10 residues. Further exploration on more complex molecular systems (*e.g.*, proteins) is warranted. Secondly, since the training labels for FBM depend on the marginal score calculations provided by FBM-BASE, we have to adopt a two-stage training process rather than an end-to-end one, which increases the training overhead. Lastly, the transitions between metastable states are still not fast enough, resulting in unreasonable conformations along the generated paths. Therefore, methods for rapid and jump-like state transition are of great importance.

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

APPENDIX

## A    REPRODUCIBILITY

Our source code and the curated dataset PepMD are all available at https://anonymous.4open.science/r/FBM-4373.

## B    PROOFS OF PROPOSITIONS

*Proof of Proposition 3.1.*    Given the joint distributions of random variables $\vec{\mathbf{X}}_0$ and $\vec{\mathbf{X}}_1$ following densities $q_0, q_1$ and $p_0, p_1$ satisfy:

$$p(\vec{\mathbf{X}}_0, \vec{\mathbf{X}}_1) = \frac{1}{Z} q(\vec{\mathbf{X}}_0, \vec{\mathbf{X}}_1) \exp(-k(\varepsilon(\vec{\mathbf{X}}_0) + \varepsilon(\vec{\mathbf{X}}_1))), \tag{18}$$

where $Z$ is the partition function to ensure $\iint p(\vec{\mathbf{X}}_0, \vec{\mathbf{X}}_1) \, d\vec{\mathbf{X}}_0 \, d\vec{\mathbf{X}}_1 = 1$. It is easy to verify that the condition holds when $\vec{\mathbf{X}}_0$ and $\vec{\mathbf{X}}_1$ are independent variables.

Therefore, the marginal density $p_t$ is given by:

$$p_t(\vec{\mathbf{X}}_t) = \iint p_t(\vec{\mathbf{X}}_t | \vec{\mathbf{X}}_0, \vec{\mathbf{X}}_1) p(\vec{\mathbf{X}}_0, \vec{\mathbf{X}}_1) \, d\vec{\mathbf{X}}_0 \, d\vec{\mathbf{X}}_1 \tag{19}$$

$$= \iint q_t(\vec{\mathbf{X}}_t | \vec{\mathbf{X}}_0, \vec{\mathbf{X}}_1) q(\vec{\mathbf{X}}_0, \vec{\mathbf{X}}_1) \frac{\exp(-k(\varepsilon(\vec{\mathbf{X}}_0) + \varepsilon(\vec{\mathbf{X}}_1)))}{Z} \, d\vec{\mathbf{X}}_0 \, d\vec{\mathbf{X}}_1 \tag{20}$$

$$= \iint q_t(\vec{\mathbf{X}}_0, \vec{\mathbf{X}}_1 | \vec{\mathbf{X}}_t) q_t(\vec{\mathbf{X}}_t) \frac{\exp(-k(\varepsilon(\vec{\mathbf{X}}_0) + \varepsilon(\vec{\mathbf{X}}_1)))}{Z} \, d\vec{\mathbf{X}}_0 \, d\vec{\mathbf{X}}_1 \tag{21}$$

$$= q_t(\vec{\mathbf{X}}_t) \mathbb{E}_{q_t(\cdot, \cdot | \vec{\mathbf{X}}_t)} \left[ \frac{\exp(-k(\varepsilon(\vec{\mathbf{X}}_0) + \varepsilon(\vec{\mathbf{X}}_1)))}{Z} \right], \tag{22}$$

where in the first equality we use the assumption that probability paths $p_t$ and $q_t$ share the same conditional distribution given $\vec{\mathbf{X}}_0, \vec{\mathbf{X}}_1$. Considering the assumption that $p_t$ admits the Boltzmann-constrained form as in Eq. (7), we can easily derive the formula for the intermediate potential:

$$\varepsilon_t(\vec{\mathbf{X}}_t) = -\frac{1}{k} \log \mathbb{E}_{q_t(\vec{\mathbf{X}}_0, \vec{\mathbf{X}}_1 | \vec{\mathbf{X}}_t)} [\exp(-k(\varepsilon(\vec{\mathbf{X}}_0) + \varepsilon(\vec{\mathbf{X}}_1)))] + \frac{1}{k} \log \frac{Z}{Z_t}. \tag{23}$$

Take the gradient of Eq. (23) with regard to $\vec{\mathbf{X}}_t$, the intermediate force field is given by:

$$\nabla \varepsilon_t(\vec{\mathbf{X}}_t) = -\frac{\iint \exp(-k(\varepsilon(\vec{\mathbf{X}}_0) + \varepsilon(\vec{\mathbf{X}}_1))) \nabla q_t(\vec{\mathbf{X}}_0, \vec{\mathbf{X}}_1 | \vec{\mathbf{X}}_t) \, d\vec{\mathbf{X}}_0 \, d\vec{\mathbf{X}}_1}{k \mathbb{E}_{q_t(\cdot, \cdot | \vec{\mathbf{X}}_t)} [\exp(-k(\varepsilon(\vec{\mathbf{X}}_0) + \varepsilon(\vec{\mathbf{X}}_1)))]}, \tag{24}$$

where we assume that the integrals and gradients can be commuted. The numerator of Eq. (24) can be further expanded by:

$$\iint \exp(-k(\varepsilon(\vec{\mathbf{X}}_0) + \varepsilon(\vec{\mathbf{X}}_1))) \nabla q_t(\vec{\mathbf{X}}_0, \vec{\mathbf{X}}_1 | \vec{\mathbf{X}}_t) \, d\vec{\mathbf{X}}_0 \, d\vec{\mathbf{X}}_1 \tag{25}$$

$$= \iint \exp(-k(\varepsilon(\vec{\mathbf{X}}_0) + \varepsilon(\vec{\mathbf{X}}_1))) q_t(\vec{\mathbf{X}}_0, \vec{\mathbf{X}}_1 | \vec{\mathbf{X}}_t) \nabla \log q_t(\vec{\mathbf{X}}_0, \vec{\mathbf{X}}_1 | \vec{\mathbf{X}}_t) \, d\vec{\mathbf{X}}_0 \, d\vec{\mathbf{X}}_1 \tag{26}$$

$$= \iint \exp(-k(\varepsilon(\vec{\mathbf{X}}_0) + \varepsilon(\vec{\mathbf{X}}_1))) q_t(\vec{\mathbf{X}}_0, \vec{\mathbf{X}}_1 | \vec{\mathbf{X}}_t) \nabla \log \frac{q_t(\vec{\mathbf{X}}_t | \vec{\mathbf{X}}_0, \vec{\mathbf{X}}_1) q(\vec{\mathbf{X}}_0, \vec{\mathbf{X}}_1)}{q_t(\vec{\mathbf{X}}_t)} \, d\vec{\mathbf{X}}_0 \, d\vec{\mathbf{X}}_1 \tag{27}$$

$$= \mathbb{E}_{q_t(\cdot, \cdot | \vec{\mathbf{X}}_t)} [\exp(-k(\varepsilon(\vec{\mathbf{X}}_0) + \varepsilon(\vec{\mathbf{X}}_1)))(\nabla \log q_t(\vec{\mathbf{X}}_t | \vec{\mathbf{X}}_0, \vec{\mathbf{X}}_1)) - \nabla \log q_t(\vec{\mathbf{X}}_t)]. \tag{28}$$

Substitute the numerator back into Eq. (24) and the conclusion holds.    $\square$

*Proof of Proposition 3.2.* Similar to the definition of $v^*(\vec{\mathbf{X}}_t, t)$ and $u^*(\vec{\mathbf{X}}_t, t)$ under the probability path $q_t$, we define $v'(\vec{\mathbf{X}}_t, t)$ and $u'(\vec{\mathbf{X}}_t, t)$ under the probability path $p_t$ accordingly, which have the following form:

$$v'(\vec{\mathbf{X}}_t, t) = \mathbb{E}_{p_t(\cdot, \cdot | \vec{\mathbf{X}}_t)}\left[\frac{\vec{\mathbf{X}}_1 - \vec{\mathbf{X}}_t}{1 - t}\right], \; u'(\vec{\mathbf{X}}_t, t) = \mathbb{E}_{p_t(\cdot, \cdot | \vec{\mathbf{X}}_t)}\left[\frac{\vec{\mathbf{X}}_t - \vec{\mathbf{X}}_0}{t}\right]. \tag{29}$$

Further, the marginal scores of probability paths $q_t, p_t$ at diffusion time $t$ (termed as $s_t^*$ and $s_t'$ respectively) are connected based on Eq. (7) as follows:

$$s_t'(\vec{\mathbf{X}}_t) = s_t^*(\vec{\mathbf{X}}_t) - k\nabla\varepsilon_t(\vec{\mathbf{X}}_t). \tag{30}$$

Considering the linkage between scores and vector fields in Eq. (12), we obtain:

$$-k\nabla\varepsilon_t(\vec{\mathbf{X}}_t) = \frac{v'(\vec{\mathbf{X}}_t, t) - u'(\vec{\mathbf{X}}_t, t)}{\sigma^2} - \frac{v^*(\vec{\mathbf{X}}_t, t) - u^*(\vec{\mathbf{X}}_t, t)}{\sigma^2} \tag{31}$$

$$= \frac{1}{\sigma^2}[(v'(\vec{\mathbf{X}}_t, t) - v^*(\vec{\mathbf{X}}_t, t)) - (u'(\vec{\mathbf{X}}_t, t) - u^*(\vec{\mathbf{X}}_t, t))]. \tag{32}$$

Now we expand the term $v' - v^*$ in the form of integral:

$$v'(\vec{\mathbf{X}}_t, t) - v^*(\vec{\mathbf{X}}_t, t) = \frac{1}{1 - t}\iint \vec{\mathbf{X}}_1\left(p_t(\vec{\mathbf{X}}_0, \vec{\mathbf{X}}_1 | \vec{\mathbf{X}}_t) - q_t(\vec{\mathbf{X}}_0, \vec{\mathbf{X}}_1 | \vec{\mathbf{X}}_t)\right)\mathrm{d}\vec{\mathbf{X}}_0\,\mathrm{d}\vec{\mathbf{X}}_1. \tag{33}$$

For convenience, we define:

$$f(\vec{\mathbf{X}}_0, \vec{\mathbf{X}}_1, \vec{\mathbf{X}}_t, t) = p_t(\vec{\mathbf{X}}_0, \vec{\mathbf{X}}_1 | \vec{\mathbf{X}}_t) - q_t(\vec{\mathbf{X}}_0, \vec{\mathbf{X}}_1 | \vec{\mathbf{X}}_t) \tag{34}$$

$$= \frac{p_t(\vec{\mathbf{X}}_t | \vec{\mathbf{X}}_0, \vec{\mathbf{X}}_1)p(\vec{\mathbf{X}}_0, \vec{\mathbf{X}}_1)}{p_t(\vec{\mathbf{X}}_t)} - \frac{q_t(\vec{\mathbf{X}}_t | \vec{\mathbf{X}}_0, \vec{\mathbf{X}}_1)q(\vec{\mathbf{X}}_0, \vec{\mathbf{X}}_1)}{q_t(\vec{\mathbf{X}}_t)} \tag{35}$$

$$= \frac{q_t(\vec{\mathbf{X}}_t | \vec{\mathbf{X}}_0, \vec{\mathbf{X}}_1)q(\vec{\mathbf{X}}_0, \vec{\mathbf{X}}_1)}{q_t(\vec{\mathbf{X}}_t)}\left[\frac{Z_t}{Z}\exp(-k(\varepsilon(\vec{\mathbf{X}}_0) + \varepsilon(\vec{\mathbf{X}}_1) - \varepsilon_t(\vec{\mathbf{X}}_t))) - 1\right], \tag{36}$$

Denote $g(\vec{\mathbf{X}}, \vec{\mathbf{X}}_t, t) = \int f(\vec{\mathbf{X}}_0, \vec{\mathbf{X}}, \vec{\mathbf{X}}_t, t)\,\mathrm{d}\vec{\mathbf{X}}_0$, Eq. (33) can be rewritten by first integrating over $\vec{\mathbf{X}}_0$:

$$v'(\vec{\mathbf{X}}_t, t) - v^*(\vec{\mathbf{X}}_t, t) = \frac{1}{1 - t}\int \vec{\mathbf{X}}g(\vec{\mathbf{X}}, \vec{\mathbf{X}}_t, t)\,\mathrm{d}\vec{\mathbf{X}}. \tag{37}$$

Similarly, we can define $h(\vec{\mathbf{X}}, \vec{\mathbf{X}}_t, t) = \int f(\vec{\mathbf{X}}, \vec{\mathbf{X}}_1, \vec{\mathbf{X}}_t, t)\,\mathrm{d}\vec{\mathbf{X}}_1$, then $u' - u^*$ admits the form:

$$u'(\vec{\mathbf{X}}_t, t) - u^*(\vec{\mathbf{X}}_t, t) = -\frac{1}{t}\int \vec{\mathbf{X}}h(\vec{\mathbf{X}}, \vec{\mathbf{X}}_t, t)\,\mathrm{d}\vec{\mathbf{X}}. \tag{38}$$

To establish the equality between $v' - v^*$ and $u' - u^*$ for solving Eq. (31), we further investigate the intrinsic relationship between functions $g$ and $h$.

Since Langevin dynamics (Langevin, 1908) in a conservative field can be considered to reach a stationary distribution after some time and satisfy the detailed balance (Bussi & Parrinello, 2007), which means $\mu(\vec{\mathbf{X}}_0)T(\vec{\mathbf{X}}_1 | \vec{\mathbf{X}}_0) = \mu(\vec{\mathbf{X}}_1)T(\vec{\mathbf{X}}_0 | \vec{\mathbf{X}}_1)$ for any states $\vec{\mathbf{X}}_0, \vec{\mathbf{X}}_1$, where $\mu(\cdot)$ denotes the equilibrium probability and $T(\cdot | \cdot)$ denotes the Markov transition probability. In particular, we assume that with a sufficiently large number of non-redundant data pairs selected from long MD trajectories, our data distribution inherits the property, namely $q(\vec{\mathbf{X}}_0, \vec{\mathbf{X}}_1) = q(\vec{\mathbf{X}}_1, \vec{\mathbf{X}}_0)$. Therefore, we have:

$$q_t(\vec{\mathbf{X}}_t) = \iint q_t(\vec{\mathbf{X}}_t | \vec{\mathbf{X}}_0, \vec{\mathbf{X}}_1)q(\vec{\mathbf{X}}_0, \vec{\mathbf{X}}_1)\,\mathrm{d}\vec{\mathbf{X}}_0\,\mathrm{d}\vec{\mathbf{X}}_1 \tag{39}$$

$$= \iint q_{1-t}(\vec{\mathbf{X}}_t | \vec{\mathbf{X}}_1, \vec{\mathbf{X}}_0)q(\vec{\mathbf{X}}_1, \vec{\mathbf{X}}_0)\,\mathrm{d}\vec{\mathbf{X}}_1\,\mathrm{d}\vec{\mathbf{X}}_0 \tag{40}$$

$$= q_{1-t}(\vec{\mathbf{X}}_t), \tag{41}$$

where in the second equality we use the property $q_t(\vec{\mathbf{X}}_t|\vec{\mathbf{X}}_0, \vec{\mathbf{X}}_1) = q_{1-t}(\vec{\mathbf{X}}_t|\vec{\mathbf{X}}_1, \vec{\mathbf{X}}_0)$, which can be verified based on Eq. (6). Similarly, we can derive $\varepsilon_t = \varepsilon_{1-t}$ from Eq. (23), thereby implying $p_t = p_{1-t}$ and $Z_t = Z_{1-t}$. Then the following equation holds based on Eq. (36):

$$f(\vec{\mathbf{X}}_0, \vec{\mathbf{X}}_1, \vec{\mathbf{X}}_t, t) = f(\vec{\mathbf{X}}_1, \vec{\mathbf{X}}_0, \vec{\mathbf{X}}_t, 1-t). \tag{42}$$

Taking all the above into consideration, an important observation is:

$$h(\vec{\mathbf{X}}, \vec{\mathbf{X}}_t, t) = g(\vec{\mathbf{X}}, \vec{\mathbf{X}}_t, 1-t). \tag{43}$$

On the other hand, when $t$ approaches 1, the limit of function $g$ is given by:

$$\lim_{t \to 1^-} g(\vec{\mathbf{X}}, \vec{\mathbf{X}}_t, t) \tag{44}$$

$$= \lim_{t \to 1^-} \frac{1}{q_t(\vec{\mathbf{X}}_t)} \int \delta(\vec{\mathbf{X}}_t - \vec{\mathbf{X}}) q(\vec{\mathbf{X}}_0, \vec{\mathbf{X}})[\frac{Z_t}{Z} \exp(-k(\varepsilon(\vec{\mathbf{X}}_0) + \varepsilon(\vec{\mathbf{X}}) - \varepsilon_t(\vec{\mathbf{X}}_t))) - 1] \, d\vec{\mathbf{X}}_0 \tag{45}$$

$$= \frac{1}{q_1(\vec{\mathbf{X}})} \int q(\vec{\mathbf{X}}_0, \vec{\mathbf{X}})[\frac{Z_1}{Z} \exp(-k\varepsilon(\vec{\mathbf{X}}_0)) - 1] \, d\vec{\mathbf{X}}_0 \tag{46}$$

$$= \frac{1}{q_1(\vec{\mathbf{X}})} Z_1 \exp(k\varepsilon(\vec{\mathbf{X}})) \int \frac{1}{Z} q(\vec{\mathbf{X}}_0, \vec{\mathbf{X}}) \exp(-k(\varepsilon(\vec{\mathbf{X}}_0) + \varepsilon(\vec{\mathbf{X}}))) \, d\vec{\mathbf{X}}_0 - 1 \tag{47}$$

$$= \frac{1}{q_1(\vec{\mathbf{X}})} Z_1 \exp(k\varepsilon(\vec{\mathbf{X}})) \int p(\vec{\mathbf{X}}_0, \vec{\mathbf{X}}) \, d\vec{\mathbf{X}}_0 - 1 \tag{48}$$

$$= \frac{1}{q_1(\vec{\mathbf{X}})} Z_1 \exp(k\varepsilon(\vec{\mathbf{X}})) p_1(\vec{\mathbf{X}}) - 1 = 0, \tag{49}$$

where in the third and fourth equality we use the relationship between the marginal distribution and the joint distribution, and in the fifth equality we apply Eq. (7).

Here we suppose the function $g$ is separable with respect to the time variable $t$. Formally, there exist functions $\iota$ and $\Gamma$ such that the following identity holds:

$$g(\vec{\mathbf{X}}, \vec{\mathbf{X}}_t, t) \equiv \iota(t)\Gamma(\vec{\mathbf{X}}, \vec{\mathbf{X}}_t). \tag{50}$$

Given $\lim_{t \to 1^-} g(\vec{\mathbf{X}}, \vec{\mathbf{X}}_t, t) = 0$, we prescribe $\iota(t) = 1 - t$ for convenience, then we have $h(\vec{\mathbf{X}}, \vec{\mathbf{X}}_t, t) = t\Gamma(\vec{\mathbf{X}}, \vec{\mathbf{X}}_t)$ and subsequently we can derive the closed-form of $v'$ based on Eq. (37) and 38 by:

$$u'(\vec{\mathbf{X}}_t, t) - u^*(\vec{\mathbf{X}}_t, t) = -\frac{1}{t} \int \vec{\mathbf{X}} h(\vec{\mathbf{X}}, \vec{\mathbf{X}}_t, t) \, d\vec{\mathbf{X}} \tag{51}$$

$$= -\int \vec{\mathbf{X}}\Gamma(\vec{\mathbf{X}}, \vec{\mathbf{X}}_t) \, d\vec{\mathbf{X}} \tag{52}$$

$$= -\frac{1}{1-t} \int \vec{\mathbf{X}}(1-t)\Gamma(\vec{\mathbf{X}}, \vec{\mathbf{X}}_t) \, d\vec{\mathbf{X}} \tag{53}$$

$$= -(v'(\vec{\mathbf{X}}_t, t) - v^*(\vec{\mathbf{X}}_t, t)), \tag{54}$$

which implies $v'(\vec{\mathbf{X}}_t, t) = v^*(\vec{\mathbf{X}}_t, t) - \frac{\sigma^2}{2}k\nabla\varepsilon_t(\vec{\mathbf{X}}_t)$ according to Eq. (31). $\qquad \square$

*Proof of Proposition 3.3.* From the definition of $s_t^*$ in Eq. (12), the following equation holds:

$$s_t^*(\vec{\mathbf{X}}_t) = \mathbb{E}_{q_t(\cdot,\cdot|\vec{\mathbf{X}}_t)}[\nabla \log q_t(\vec{\mathbf{X}}_t|\vec{\mathbf{X}}_0, \vec{\mathbf{X}}_1)] \tag{55}$$

$$= \iint \nabla \log q_t(\vec{\mathbf{X}}_t|\vec{\mathbf{X}}_0, \vec{\mathbf{X}}_1) q_t(\vec{\mathbf{X}}_0, \vec{\mathbf{X}}_1|\vec{\mathbf{X}}_t) \, d\vec{\mathbf{X}}_0 \, d\vec{\mathbf{X}}_1 \tag{56}$$

$$= \frac{1}{q_t(\vec{\mathbf{X}}_t)} \iint \nabla q_t(\vec{\mathbf{X}}_t|\vec{\mathbf{X}}_0, \vec{\mathbf{X}}_1) q(\vec{\mathbf{X}}_0, \vec{\mathbf{X}}_1) \, d\vec{\mathbf{X}}_0 \, d\vec{\mathbf{X}}_1 \tag{57}$$

$$= \frac{1}{q_t(\vec{\mathbf{X}}_t)} \nabla \iint q_t(\vec{\mathbf{X}}_t|\vec{\mathbf{X}}_0, \vec{\mathbf{X}}_1) q(\vec{\mathbf{X}}_0, \vec{\mathbf{X}}_1) \, d\vec{\mathbf{X}}_0 \, d\vec{\mathbf{X}}_1 \tag{58}$$

$$= \frac{\nabla q_t(\vec{\mathbf{X}}_t)}{q_t(\vec{\mathbf{X}}_t)} = \nabla \log q_t(\vec{\mathbf{X}}_t), \tag{59}$$

where in the second equality we use the Bayesian rule of probability, $q(\vec{\mathbf{X}}_0, \vec{\mathbf{X}}_1)$ denotes the joint distribution of random variables $\vec{\mathbf{X}}_0, \vec{\mathbf{X}}_1$ following $q_0, q_1$. Furthermore, the third equality is justified by assuming the integrands satisfy the regularity conditions of the Leibniz Rule. $\qquad\square$

*Proof of Proposition 3.4.* We first check whether the continuity condition holds when $t \to 0^+$. Note that under this condition, the density $q_t(\vec{\mathbf{X}}_0, \vec{\mathbf{X}}_1 | \vec{\mathbf{X}}_t)$ involves into the Dirac mass $\delta(\vec{\mathbf{X}}_t - \vec{\mathbf{X}}_0)$ at point $\vec{\mathbf{X}}_t$, subsequently we have:

$$\lim_{t \to 0^+} \nabla \varepsilon_t(\vec{\mathbf{X}}_t) = -\lim_{t \to 0^+} \frac{\iint \exp(-k(\varepsilon(\vec{\mathbf{X}}_0) + \varepsilon(\vec{\mathbf{X}}_1)))\delta(\vec{\mathbf{X}}_t - \vec{\mathbf{X}}_0)\, \mathrm{d}\vec{\mathbf{X}}_0\, \mathrm{d}\vec{\mathbf{X}}_1}{k \iint \exp(-k(\varepsilon(\vec{\mathbf{X}}_0) + \varepsilon(\vec{\mathbf{X}}_1)))\delta(\vec{\mathbf{X}}_t - \vec{\mathbf{X}}_0)\, \mathrm{d}\vec{\mathbf{X}}_0\, \mathrm{d}\vec{\mathbf{X}}_1} \tag{60}$$

$$= -\lim_{t \to 0^+} \frac{\nabla \iint \exp(-k(\varepsilon(\vec{\mathbf{X}}_0) + \varepsilon(\vec{\mathbf{X}}_1)))\delta(\vec{\mathbf{X}}_t - \vec{\mathbf{X}}_0)\, \mathrm{d}\vec{\mathbf{X}}_0\, \mathrm{d}\vec{\mathbf{X}}_1}{k \int \exp(-k(\varepsilon(\vec{\mathbf{X}}_t) + \varepsilon(\vec{\mathbf{X}}_1)))\, \mathrm{d}\vec{\mathbf{X}}_1} \tag{61}$$

$$= -\lim_{t \to 0^+} \frac{\nabla \int \exp(-k(\varepsilon(\vec{\mathbf{X}}_t) + \varepsilon(\vec{\mathbf{X}}_1)))\, \mathrm{d}\vec{\mathbf{X}}_1}{k \int \exp(-k(\varepsilon(\vec{\mathbf{X}}_t) + \varepsilon(\vec{\mathbf{X}}_1)))\, \mathrm{d}\vec{\mathbf{X}}_1} \tag{62}$$

$$= -\lim_{t \to 0^+} \frac{\nabla \exp(-k\varepsilon(\vec{\mathbf{X}}_t))}{k \exp(-k\varepsilon(\vec{\mathbf{X}}_t))} = \nabla\varepsilon(\vec{\mathbf{X}}_0). \tag{63}$$

The case when $t$ approaches 1 is completely symmetrical and will not be elaborated further. Thus we have proven that the intermediate force field converges to the MD force field when $t$ approaches 0 and 1.

## C  MODEL ARCHITECTURE

In this work, we leverage the powerful `TorchMD-NET` (Pelaez et al., 2024) as the backbone model to process molecular graphs, which intrinsically satisfies $\mathrm{SO}(3)$-equivariance with the *equivariant transformer* (Thölke & Fabritiis, 2022) component. To adapt to our task setup, the inputs include not only the Cartesian coordinates $\vec{X}$ and atom embeddings $Z$, but also a one-dimensional continuous diffusion time $t$. `TorchMD-NET` will then output $\mathrm{SO}(3)$-equivariant vectors $\vec{V} \in \mathbb{R}^{N \times 3 \times H}$ and node representations $H \in \mathbb{R}^{N \times H}$. Formally, we have:

$$\vec{V}, H = \texttt{TorchMD-NET}(\vec{X}, Z, t). \tag{64}$$

To streamline the model, we only add lightweight output heads to a single `TorchMD-NET` module for the baseline model as well as FBM. For the baseline model, we use two separate two-layer Feed-Forward Networks (FFN) with no shared weights to transform the node representations into weights of vectors, which are then multiplied by $\vec{V}$ to obtain the final representation:

$$v_\theta(\vec{X}, t), u_\theta(\vec{X}, t) := \vec{V} \times \mathrm{FFN}(H) \in \mathbb{R}^{N \times 3}, \tag{65}$$

where the dimensions of hidden and output layers of FFN are all $H$ and we use SiLU (Dong et al., 2017) for activation layers. Further, we construct the networks $\alpha_\theta, \beta_\theta, \gamma_\theta$ of FBM in the same way, while the only difference is that we add one LayerNorm (Ba et al., 2016) before the FFN layer due to the variance in scale of different targets.

## D  TRAINING AND INFERENCE DETAILS

In this section, we provide additional details and pseudo codes for training and inference of the baseline bridge matching model FBM-BASE and the force-guided bridge matching model FBM.

### D.1  NORMALIZATION OF ENERGIES AND FORCES

In practice, we found that the unnormalized potential function is numerically unstable and its variance is positively correlated with the number of atoms $N$. For stable training, we need to perform certain pre-processing steps. Specifically, for potentials in `kJ/mol`, we divide by $3N$ to obtain the average potential energy per degree of freedom for the entire molecule, where $N$ varies with different peptides. For force fields in `kJ/(mol·nm)`, due to their relatively stable values across different molecular systems, we empirically multiply by the constant 0.002 for normalization.

## D.2 GUIDANCE STRENGTH $\eta$

Similar to Wang et al. (2024), we introduce the *guidance strength* $\eta$ for better approximation of the Boltzmann distribution. Formally, for any positive constant $\eta > 0$, we can define a new probability path based on $q_t$:

$$p_t(\vec{\mathbf{X}}_t) = \frac{1}{Z_t} q_t(\vec{\mathbf{X}}_t) \exp(-\frac{2\eta}{\sigma^2} k\varepsilon_t(\vec{\mathbf{X}}_t)), \varepsilon_0 = \varepsilon_1 = \varepsilon. \tag{66}$$

The only difference with Eq. (7) is the constant $2\eta/\sigma^2$ in the exponential term, which can be interpreted as how well the probability path $p_t$ is guided by energies and forces. According to Proposition 3.2, it can be easily deduced that the vector field $v'(\vec{\mathbf{X}}_t, t)$ which generates $p_t$ has the following form:

$$v'(\vec{\mathbf{X}}_t, t) = v^*(\vec{\mathbf{X}}_t, t) - \eta \cdot k\varepsilon_t(\vec{\mathbf{X}}_t). \tag{67}$$

Thus practically, we regard $\eta$ as a hyperparameter during inference and enhance the similarity between $p_t$ and the Boltzmann distribution by selecting the proper guidance strength $\eta$.

## D.3 REFINEMENT WITH CONSTRAINED ENERGY MINIMIZATION

We utilize the discrete form of the SDE process in Eq. (4) for inference with $T$ SDE steps, and the full conformation ensembles are generated in an autoregressive way, where the output from the previous step serves as the input for the next step. However, in the autoregressive fashion, errors at each inference step will accumulate, leading to out-of-distribution problem. Here we introduce an additional energy minimization procedure using `OpenMM` (Eastman et al., 2017) for refinement, which is performed for each generated conformation before sent to the next inference step. Note that we aim for the refinement to affect only the minor details (*e.g.*, X-H bonds) without altering the overall conformation; therefore, independent harmonic constraints are further applied on all heavy atoms with spring constant of 10 kcal/mol·Å$^2$ and the tolerance of 2.39 kcal/mol·Å$^2$ without maximal step limits (Wang et al., 2024).

## D.4 ALGORITHMS FOR TRAINING AND INFERENCE

We provide pseudo codes for training and inference with our models FBM-BASE and FBM in Algorithm 1,2,3 respectively.

---

**Algorithm 1** Training with FBM-BASE

---

1: **Input:** peptide pairs $(\mathcal{G}_0, \mathcal{G}_1)$ in a batch $B$, vector field networks $u(\vec{\mathbf{X}}_t, t), v(\vec{\mathbf{X}}_t, t)$
2: **for** training iterations **do**
3:     $t \sim \text{Uni}(0, 1)$
4:     $\vec{\boldsymbol{X}}_t \sim q_t(\vec{\boldsymbol{X}}_t | \vec{\boldsymbol{X}}_0, \vec{\boldsymbol{X}}_1)$
5:     $\hat{\vec{\boldsymbol{X}}}_0 \leftarrow \vec{\boldsymbol{X}}_t - t u_\theta(\vec{\boldsymbol{X}}_t, t), \hat{\vec{\boldsymbol{X}}}_1 \leftarrow \vec{\boldsymbol{X}}_t + (1-t)v_\theta(\vec{\boldsymbol{X}}_t, t)$
6:     $(\boldsymbol{D}_0, \boldsymbol{D}_1, \hat{\boldsymbol{D}}_0, \hat{\boldsymbol{D}}_1) \leftarrow$ pairwise interatomic distances of $(\vec{\boldsymbol{X}}_0, \vec{\boldsymbol{X}}_1, \hat{\vec{\boldsymbol{X}}}_0, \hat{\vec{\boldsymbol{X}}}_1)$
7:     $\mathcal{L}_{\text{fwd}} \leftarrow \frac{1}{B} \sum_{\mathcal{G}_0, \mathcal{G}_1} ||(\vec{\boldsymbol{X}}_1 - \vec{\boldsymbol{X}}_t)/(1-t) - v_\theta(\vec{\boldsymbol{X}}_t, t)||^2$
8:     $\mathcal{L}_{\text{rev}} \leftarrow \frac{1}{B} \sum_{\mathcal{G}_0, \mathcal{G}_1} ||(\vec{\boldsymbol{X}}_t - \vec{\boldsymbol{X}}_0)/t - u_\theta(\vec{\boldsymbol{X}}_t, t)||^2$
9:     $\mathcal{L}_{\text{aux}} \leftarrow \frac{1}{B} \sum_{\mathcal{G}_0, \mathcal{G}_1}(1-t) \cdot \frac{||\mathbf{1}_{D_0 < 6\text{Å}}(\boldsymbol{D}_0 - \hat{\boldsymbol{D}}_0)||^2}{\sum \mathbf{1}_{D_0 < 6\text{Å}} - N} + t \cdot \frac{||\mathbf{1}_{D_1 < 6\text{Å}}(\boldsymbol{D}_1 - \hat{\boldsymbol{D}}_1)||^2}{\sum \mathbf{1}_{D_1 < 6\text{Å}} - N}$
10:     $\mathcal{L}_{\text{base}} \leftarrow \mathcal{L}_{\text{fwd}} + \mathcal{L}_{\text{rev}} + 0.25 \cdot \mathcal{L}_{\text{aux}}$
11:     $\min \mathcal{L}_{\text{base}}$
12: **end for**

---

## D.5 HYPERPARAMETERS

The hyperparameters we choose are listed in Table 3.

---

**Algorithm 2** Training with FBM

---

1: **Input:** peptide pairs $(\mathcal{G}_0, \mathcal{G}_1)$ of **one molecular system** in a batch $B$, baseline model $v_\theta(\vec{\mathbf{X}}_t, t), u_\theta(\vec{\mathbf{X}}_t, t)$ in § 3.2 with frozen parameters, MD potentials $\varepsilon(\vec{\mathbf{X}}_0), \varepsilon(\vec{\mathbf{X}}_1)$, MD force fields $\nabla\varepsilon(\vec{\mathbf{X}}_0), \nabla\varepsilon(\vec{\mathbf{X}}_1)$, force field networks $\alpha_\theta(\vec{\mathbf{X}}_t, t), \beta_\theta(\vec{\mathbf{X}}_t, t), \gamma_\theta(\vec{\mathbf{X}}_t, t)$
2: **for** training iterations **do**
3:   $t \sim \mathrm{Uni}(0, 1)$
4:   $\vec{\mathbf{X}}_t \sim q_t(\vec{\mathbf{X}}_t | \vec{\mathbf{X}}_0, \vec{\mathbf{X}}_1)$
5:   $s_t^*(\vec{\mathbf{X}}_t) \leftarrow (v_\theta(\vec{\mathbf{X}}_t, t) - u_\theta(\vec{\mathbf{X}}_t, t))/\sigma^2$
6:   $\zeta(\vec{\mathbf{X}}_0, \vec{\mathbf{X}}_1, \vec{\mathbf{X}}_t) \leftarrow s_t^*(\vec{\mathbf{X}}_t) - \nabla \log q_t(\vec{\mathbf{X}}_t | \vec{\mathbf{X}}_0, \vec{\mathbf{X}}_1)$
7:   $M \leftarrow \frac{1}{B} \sum_{\mathcal{G}_0, \mathcal{G}_1} q_t(\vec{\mathbf{X}}_t | \vec{\mathbf{X}}_0, \vec{\mathbf{X}}_1) \exp(-k(\varepsilon(\vec{\mathbf{X}}_0) + \varepsilon(\vec{\mathbf{X}}_1)))$
8:   $w(\vec{\mathbf{X}}_t, t) \leftarrow (1 - t) \cdot \mathtt{detach}(\alpha_\theta(\vec{\mathbf{X}}_t, t)) + t \cdot \mathtt{detach}(\beta_\theta(\vec{\mathbf{X}}_t, t)) + t(1 - t) \cdot \gamma_\theta(\vec{\mathbf{X}}_t, t)$
9:   $\mathcal{L}_{\mathrm{iff}} \leftarrow \frac{1}{B} \sum_{\mathcal{G}_0, \mathcal{G}_1} ||\exp(-k(\varepsilon(\vec{\mathbf{X}}_0) + \varepsilon(\vec{\mathbf{X}}_1)))\zeta(\vec{\mathbf{X}}_0, \vec{\mathbf{X}}_1, \vec{\mathbf{X}}_t)/kM - w_\theta(\vec{\mathbf{X}}_t, t)||^2$
10:   $\mathcal{L}_{\mathrm{bnd}} \leftarrow \frac{1}{B} \sum_{\mathcal{G}_0, \mathcal{G}_1} ||\nabla\varepsilon(\vec{\mathbf{X}}_0) - \alpha_\theta(\vec{\mathbf{X}}_t, t)||^2 + ||\nabla\varepsilon(\vec{\mathbf{X}}_1) - \beta_\theta(\vec{\mathbf{X}}_t, t)||^2$
11:   $\mathcal{L}_{\mathrm{FBM}} \leftarrow \mathcal{L}_{\mathrm{iff}} + \mathcal{L}_{\mathrm{bnd}}$
12:   $\min \mathcal{L}_{\mathrm{FBM}}$
13: **end for**

---

**Algorithm 3** Autoregressive inference with FBM/FBM-BASE

---

1: **Input:** Initial state $\mathcal{G}_0$, chain length $L$, discrete SDE step $T$, guidance strength $\eta$, baseline model $v(\vec{\mathbf{X}}_t, t)$ in § 3.2, FBM model $w(\vec{\mathbf{X}}_t, t)$ in § 3.3, model type $c \in \{\text{FBM-BASE, FBM}\}$
2: $C \leftarrow []$
3: $\Delta \leftarrow 1/T$
4: **for** $l \leftarrow 1$ **to** $L$ **do**
5:   **for** $t$ **in** $\mathrm{linspace}(0, 1 - \Delta, T)$ **do**
6:     $\epsilon \sim \mathcal{N}(\mathbf{0}, \boldsymbol{I})$
7:     **if** $c = \text{FBM}$ **then**
8:       $v'(\vec{\mathbf{X}}_t, t) \leftarrow v_\theta(\vec{\mathbf{X}}_t, t) - \eta \cdot kw_\theta(\vec{\mathbf{X}}_t, t)$
9:     **else**
10:       $v'(\vec{\mathbf{X}}_t, t) \leftarrow v_\theta(\vec{\mathbf{X}}_t, t)$
11:     **end if**
12:     $\vec{\mathbf{X}}_{t+\Delta} \leftarrow \vec{\mathbf{X}}_t + v'(\vec{\mathbf{X}}_t, t)\Delta + \sqrt{t}\sigma\epsilon$
13:   **end for**
14:   $\vec{\mathbf{X}}_1' \leftarrow \mathrm{energy\_minim}(\vec{\mathbf{X}}_1)$
15:   $\vec{\mathbf{X}}_0 \leftarrow \vec{\mathbf{X}}_1'$
16:   $C \leftarrow C \cup \vec{\mathbf{X}}_1$
17: **end for**
18: **Output** $C$

---

# E EXPERIMENTAL DETAILS

## E.1 DATASET DETAILS

As mentioned in § 4, all peptides of PepMD are simulated using `OpenMM` (Eastman et al., 2017). The parameters we used for MD simulations are listed in Table 4 and the statistical information of PepMD is shown in Table 5.

Additionally, all 14 peptides of our test set are listed below with the format {pdb-id}:{chain-id}, including 1hhg:C, 1k8d:P, 1k83:M, 1bz9:C, 1i7u:C, 1gxc:B, 1ar8:0, 2xa7:P, 1e28:C, 1gy3:F, 1n73:I, 1fpr:B, 1aze:B, 1qj6:I.

## E.2 DETAILS ON EVALUATION METRICS

In this part, we provide details for computing the evaluation metrics in § 4.

Table 3: Hyperparameter choice of FBM-BASE and FBM.

| Hyperparameters | Values |
|---|---|
| **Network** | |
| Hidden dimension $H$ of FBM-BASE | 128 |
| Hidden dimension $H$ of FBM | 176 |
| RBF dimension | 32 |
| Number of attention heads | 8 |
| Number of layers | 6 |
| Cutoff threshold $r_{\text{cut}}$ | 5.0Å |
| **Training** | |
| Learning rate | 5e-4 |
| Optimizer | Adam |
| Warm up steps | 1,000 |
| Warm up scheduler | LamdaLR |
| Training scheduler | ReduceLRonPlateau(factor=0.8, patience=5, min_lr=1e-7) |
| Batch size of FBM-BASE | 16 |
| Batch size of FBM | 10 |
| SDE noise scale $\sigma$ | 0.2 |
| **Inference** | |
| SDE steps $T$ | [25,30] |
| Guidance strength $\eta$ of FBM | [0.04,0.05,0.06,0.07,0.08] |

Table 4: MD simulation setups using `OpenMM`.

| Property | Value |
|---|---|
| Forcefield | AMBER-14 |
| Integrator | LangevinMiddleIntegrator |
| Integration time step | 1fs |
| Frame spacing | 1ps |
| Friction coefficient | $1.0\text{ps}^{-1}$ |
| Temperature | 300K |
| Electrostatics | NoCutoff |
| Constraints | HBonds |

**Flexibility** Following Janson et al. (2023), we compute the contact rates between residues as a measure of structural flexibility. For each residue pair $i, j$ ($1 \leq i < j \leq R$) of a peptide with $R$ residues, the contact rate $r(i, j)$ of residue $i, j$ is defined as follows:

$$r(i,j) = \frac{1}{L} \sum_{l=1}^{L} \mathbf{1}_{d_l(i,j) < 10\text{Å}}, \tag{68}$$

where $d_l(i, j)$ denotes the Euclidean distance between $\alpha$-carbons of residue $i, j$ of conformation $l$. Now we compute the root mean square error of contact maps between generated conformation ensembles and reference MD trajectories:

$$\text{CONTACT} = \sqrt{\frac{2}{R(R-1)} \sum_{1 \leq i < j \leq R} (r(i,j) - r_{\text{ref}}(i,j))^2}. \tag{69}$$

**Validity** We assess the structural validity by checking for bond breaks between adjacent residues and bond clashes between any residue pairs. The same as in Wang et al. (2024), *bond clash* occurs

Table 5: Dataset statistics.

| Dataset name | PepMD |
|---|---|
| Training set simulation time | 100ns |
| Test set simulation time | 100ns |
| MD integration time step $\Delta t$ | 1fs |
| coarsened predition time $\tau$ | $0.5 \times 10^6$fs |
| # Clusters | 2480 |
| # Training peptides | 136 |
| # Training pairs per peptide | $2 \times 10^3$ |
| # Validation pairs per peptide | $4 \times 10^2$ |
| # Test peptides | 14 |

when the distance between $\alpha$-carbons of any residue pair is less than the threshold $\delta_{\text{clash}} = 3.0\text{Å}$, and *bond break* occurs when the distance between adjacent $\alpha$-carbons is greater than the threshold $\delta_{\text{break}} = 4.19\text{Å}$. Then the metric VAL-CA is assessed by the fraction of conformations without bond break and bond clash.

**Distributional Similarity**   Similar to Lu et al. (2023b), we project peptide conformations onto the following three low-dimensional feature space: (i) Pairwise Distance (PWD) between $\alpha$-carbons excluding residue pairs within an offset of 3. (ii) Radius of gyration (RG) which computes the geometric mean of the distances from $\alpha$-carbons to the center-of-mass. (iii) Time-lagged Independent Components (TIC), where we featurize structures using backbone dihedrals $\psi, \phi, \omega$ and pairwise distances between $\alpha$-carbons (Klein et al., 2024), then TIC analysis is performed using `Deeptime` (Hoffmann et al., 2021). Only the slowest components, TIC 0 and TIC 1, are taken for further evaluation (Pérez-Hernández et al., 2013). (iv) the joint distribution of TIC 0 and TIC 1, termed as TIC-2D. (v) Specifically for the evaluation on AD, the joint distribution of backbone dihedrals $\psi$ and $\phi$, namely the *Ramachandran plot* (Ramachandran et al., 1963), is taken into consideration (RAM).

Afterwards we compute the Jensen-Shannon (JS) distance between generated samples and reference MD trajectories on the projection space. Features are discretized with 50 bins based on the reference ensembles, and a pseudo count 1e-6 is added for numerical stability. For each feature space, we report the mean distance along all dimensions.

# F   ADDITIONAL EXPERIMENTAL RESULTS

## F.1   ABLATION STUDY

In Table 6 we provide ablation results of SDE steps $T$ and the guidance strength $\eta$ on test peptides of PepMD. A clear pattern is that when $T$ is fixed, the greater the guidance strength $\eta$, the more likely it is to generate reasonable conformations, which demonstrates a strong correlation between the intermediate force field and real interatomic constraints of molecular systems.

Table 6: Ablation results of SDE steps $T$ and the guidance strength $\eta$ on the test set of PepMD. Values of each metric are first averaged over 3 independent runs for each peptide and then shown in mean/std of all 14 test peptides.

| Hyperparameters | JS DISTANCE ($\downarrow$) | | | | VAL-CA ($\uparrow$) | CONTACT ($\downarrow$) |
| --- | --- | --- | --- | --- | --- | --- |
| | PWD | RG | TIC | TIC-2D | | |
| $T = 25, \eta = 0.04$ | 0.586/0.059 | 0.540/0.143 | 0.640/0.056 | 0.809/0.020 | 0.559/0.190 | 0.195/0.115 |
| $T = 25, \eta = 0.05$ | 0.578/0.069 | 0.550/0.151 | 0.639/0.075 | 0.805/0.027 | 0.606/0.192 | 0.179/0.132 |
| $T = 25, \eta = 0.06$ | 0.573/0.064 | 0.542/0.140 | 0.631/0.077 | 0.801/0.032 | 0.616/0.188 | 0.188/0.127 |
| $T = 25, \eta = 0.07$ | 0.585/0.084 | 0.548/0.177 | 0.644/0.089 | 0.800/0.044 | 0.647/0.182 | 0.201/0.135 |
| $T = 25, \eta = 0.08$ | 0.573/0.064 | 0.577/0.144 | 0.638/0.079 | 0.804/0.033 | 0.675/0.169 | 0.200/0.111 |
| $T = 30, \eta = 0.04$ | 0.585/0.074 | 0.587/0.151 | 0.644/0.077 | 0.804/0.029 | 0.553/0.208 | 0.208/0.131 |
| $T = 30, \eta = 0.05$ | 0.582/0.066 | 0.542/0.162 | 0.640/0.088 | 0.806/0.035 | 0.615/0.183 | 0.217/0.130 |
| $T = 30, \eta = 0.06$ | 0.596/0.085 | 0.591/0.139 | 0.637/0.089 | 0.803/0.041 | 0.604/0.205 | 0.231/0.116 |
| $T = 30, \eta = 0.07$ | 0.576/0.059 | 0.590/0.120 | 0.649/0.074 | 0.806/0.023 | 0.655/0.180 | 0.182/0.116 |
| $T = 30, \eta = 0.08$ | 0.590/0.077 | 0.575/0.151 | 0.614/0.101 | 0.789/0.065 | 0.661/0.176 | 0.208/0.116 |

## F.2 PEPMD ADDITIONAL RESULTS

In Figure 6, we provide the comparison of different methods for free energy projections on the slowest two TIC components for test peptides 1n73:I (GHRP), 1gxc:B (RHFDTYLIRR) and 1qj6:I (DFEEIPEEYL) from PepMD. Note that, due to the discrepancies in the prediction time interval and the number of samples, there may be certain systematic errors compared with the MD simulation data. Compared to the baselines, FBM presents more accurate depictions of the molecular free energy landscape in most cases. We also found that for peptides with fewer residues (*e.g.*, the tetrapeptide 1n73:I), FBM often achieves higher accuracy. This condition aligns with the scaling law, suggesting that accurate molecular dynamics simulations for more complex molecules may require larger training datasets and more parameters.

To provide a more comprehensive evaluation, we present the visualization of comprehensive metrics for the three peptides in Figure 7 similar to that in Figure 4. Note that, the PWD distribution plot of peptide 1n73:I is not displayed since its residue length is too short for an offset of 3 (Wang et al., 2024). FBM undoubtedly outperforms all other baselines in stably generating valid conformations of all three peptides. For the tetrapeptide 1n73:I and decapeptide 1gxc:B, FBM achieves a close match with MD trajectories in terms of distributions on projected features, residue contact rates and inter-residue distances. Thrillingly, FBM consistently achieves equilibrium distributions during the inference of both peptides and aligns quite well with MD trajectories under the $C_\alpha$-RMSD evaluation, showing a good transferability to Out-Of-Distribution (OOD) peptides with different residue lengths. In contrast, other methods either deviate from the real distribution or exhibit excessive fluctuations during the generation process.

Meanwhile, we also provide a **failure case** of FBM, *i.e.*, peptide 1qj6:I in Figure 7(c). For this peptide, the trajectories generated by FBM show significant deviation from the reference distribution. Based on the residue contact map, we observe that the residues in the peptide are spatially dispersed, which may hinder the graph neural network from efficiently capturing global dynamics and interactions of the molecule. In such cases, increasing the model parameter size, stacking more layers, and expanding the dataset are likely to help generate more accurate trajectories.

## F.3 A TINY EXPERIMENT ON CHIGNOLIN

To extend FBM on more complex molecular systems, we perform a tiny experiment on Chignolin, a small protein consisting of 10 residues and 175 atoms. The trajectory data of Chignolin is downloaded from `figshare`[3], which was curated by Culubret & Fabritiis (2021). MD simulations were performed with ACEMD, using CHARMM22 force field (MacKerell Jr et al., 1998) and TIP3P water model (Jorgensen et al., 1983) at 350K temperature, which contains 1,881 water molecules and two $Na^+$ ions to neutralize the peptide's negative charge. The dataset consists of 3,744 independent

---

[3]https://figshare.com/articles/dataset/Chignolin_Simulations/13858898

product simulations of 50 ns, for a total aggregate time of 187.2 μs. Considering the difference of MD simulation setups from ours, we need to fine-tune our model on the dataset to align with its simulation environment.

To be specific, we first randomly select 500/50/3 **independent** trajectories from the dataset for training/validation/test and select 200 data pairs for each trajectory with $\tau =$100 ps. The trained FBM-BASE model will then be fine-tuned on the training data for 20 epochs, with a relatively low learning rate of 2e-4. Further, to obtain atomic forces and potentials of molecules for training with FBM, we use CHARMM36 force field (Best et al., 2012) and implicit solvation of GB-OBC I parameters (Onufriev et al., 2004) on the `OpenMM` platform. Afterwards, a new FBM model will be trained for 100 epochs, with the fine-tuned FBM-BASE as the baseline model.

The evaluation results on three test trajectories are shown in Table 7, where the identifiers of these trajectories in the original dataset are labeled as e1s44, e59s7 and e3s24, respectively. We find that FBM significantly outperforms FBM-BASE across multiple comprehensive metrics, showcasing a strong and stable generation ability for molecular dynamics. Moreover, we provide the visualizations of various metrics on the test set in Figure 8, where the generated trajectories of FBM shows a close match to those of MD in most cases, demonstrating its usefulness and scalability to more complex molecular systems.

Table 7: Evaluation results of FBM on three test trajectories of Chignolin. Values of each metric are first averaged over 3 independent runs for each peptide and then shown in mean/std of all 14 test peptides. The best result for each metric is shown in **bold** and the second best is underlined.

| Index | Model | JS DISTANCE (↓) | | | | VAL-CA (↑) | CONTACT (↓) |
|-------|-------|------|------|------|------|------------|-------------|
| | | PWD | RG | TIC | RAM | | |
| e1s44 | FBM | **0.315**/0.056 | **0.285**/0.080 | **0.544**/0.011 | 0.509/0.029 | **0.691**/0.041 | **0.161**/0.069 |
| | FBM-BASE | 0.417/0.069 | 0.467/0.148 | 0.554/0.023 | **0.480**/0.013 | 0.465/0.057 | 0.249/0.066 |
| e59s7 | FBM | **0.395**/0.017 | **0.400**/0.034 | **0.522**/0.015 | **0.443**/0.020 | **0.780**/0.012 | **0.184**/0.029 |
| | FBM-BASE | 0.456/0.026 | 0.407/0.016 | 0.526/0.002 | 0.490/0.017 | 0.460/0.031 | 0.219/0.056 |
| e3s24 | FBM | **0.305**/0.046 | **0.332**/0.089 | **0.524**/0.004 | 0.519/0.011 | **0.628**/0.010 | **0.123**/0.032 |
| | FBM-BASE | 0.450/0.086 | 0.549/0.149 | 0.527/0.015 | **0.502**/0.019 | 0.490/0.040 | 0.278/0.104 |

# G  COMPUTING INFRASTRUCTURE

Our models, FBM-BASE and FBM, were trained on 4 NVIDIA GeForce RTX 3090 GPUs within a week. The inference procedure with baselines and our model were all performed on one single NVIDIA GeForce RTX 3090 GPU.

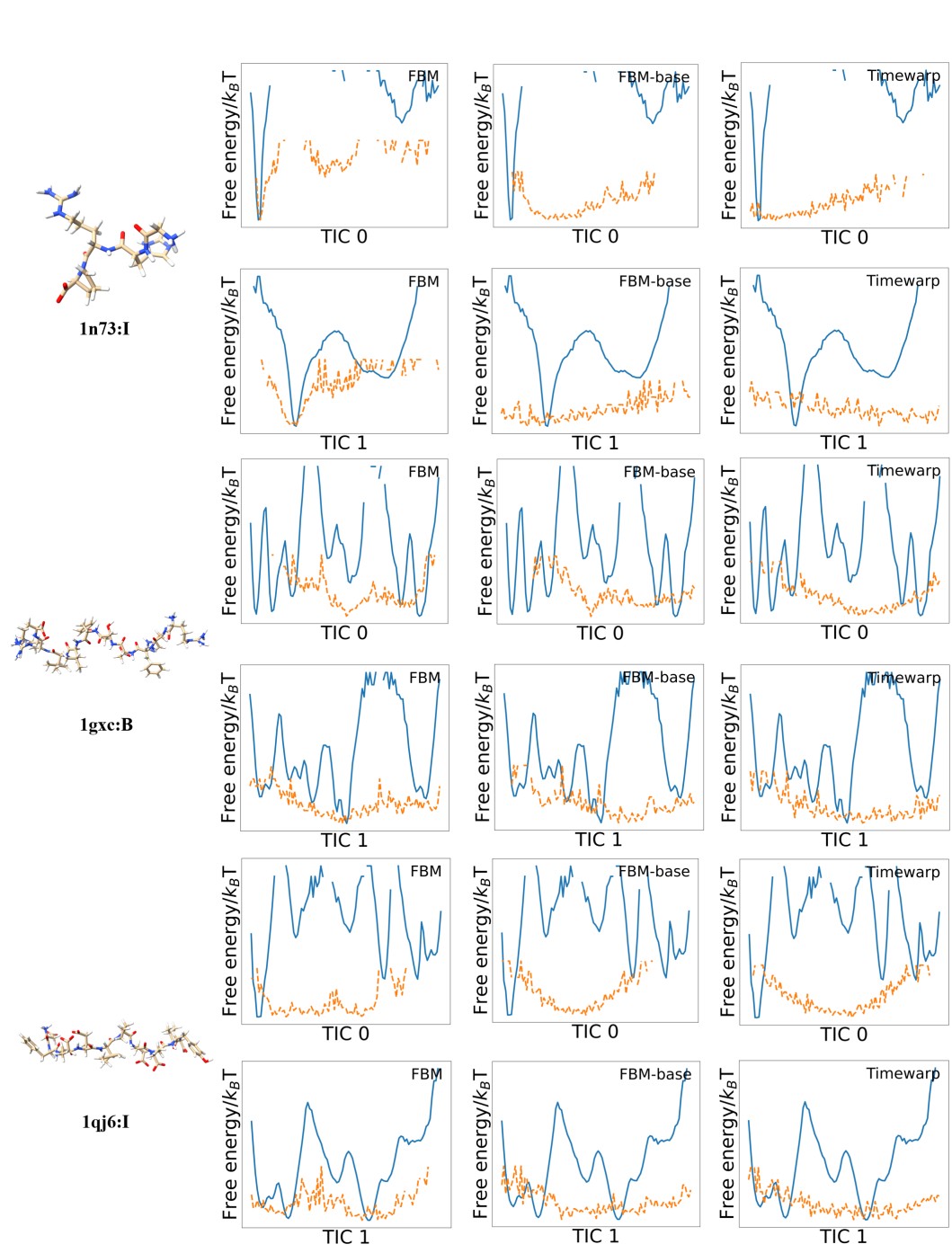

Figure 6: Experiments on PepMD test peptides 1n73:I, 1gxc:B and 1qj6:I (top, middle and bottom). Samples were generated in the time-coarsened manner for a chain length of $10^3$. Free energies (*i.e.*, the relative log probability) along the first two TIC components of FBM, FBM-BASE and Timewarp are displayed in the left, middle, and right columns, respectively. The blue solid line represents the full MD trajectories, while the yellow dashed line represents model-generated samples.

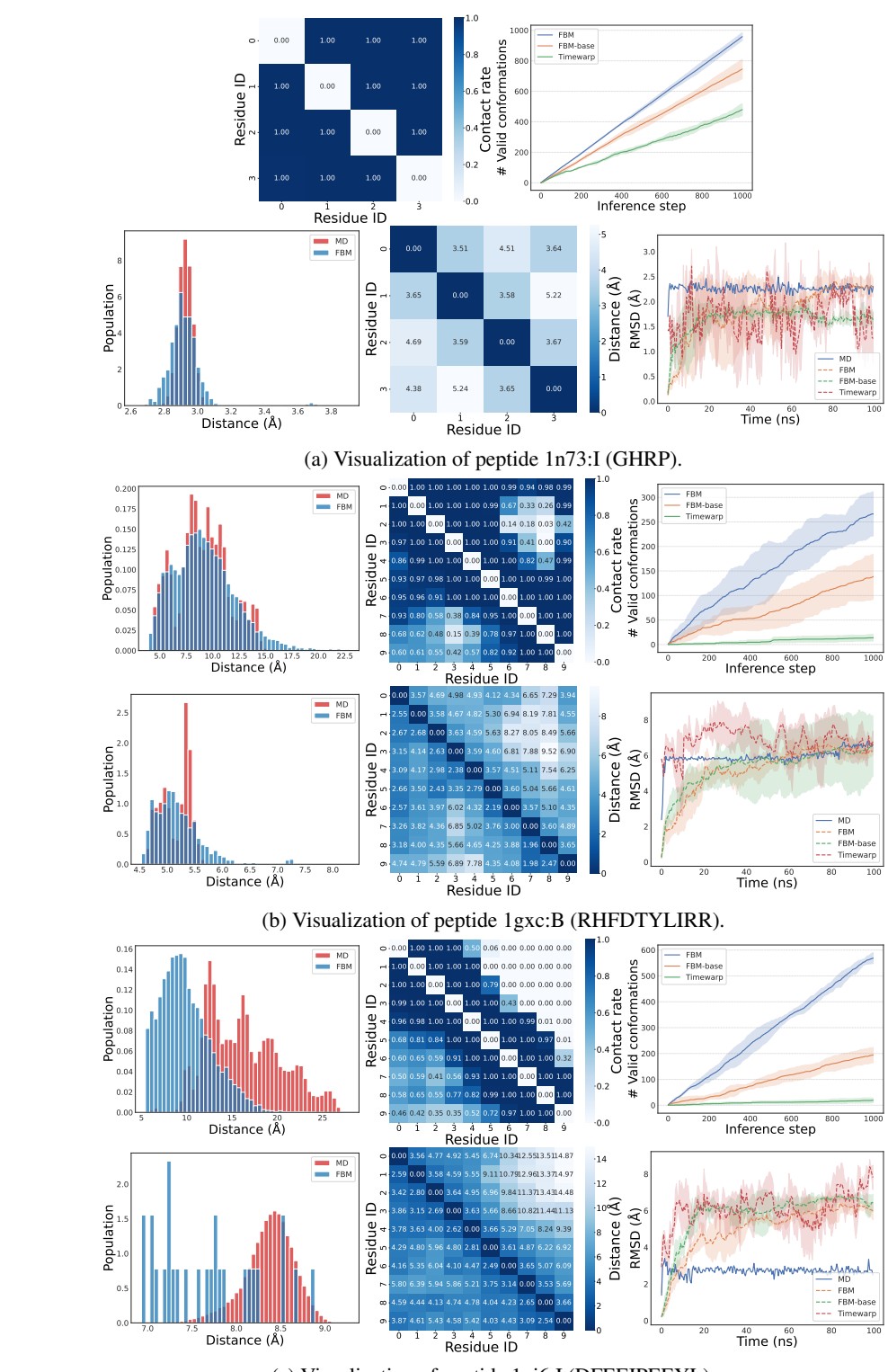

(a) Visualization of peptide 1n73:I (GHRP).

(b) Visualization of peptide 1gxc:B (RHFDTYLIRR).

(c) Visualization of peptide 1qj6:I (DFEEIPEEYL).

Figure 7: The visualization of comprehensive metrics on test peptides 1n73:I, 1gxc:B and 1qj6:I (top, middle and bottom). For each sub-figure of the corresponding peptide: **1.** The top-left and bottom-left plots show the joint distribution of pairwise distances between residues and the distribution of the radius of gyration, respectively. **2.** The top-middle and bottom-middle plots demonstrate the residue contact map and residue minimum-distance map, respectively. **3.** The top-right plot compares the cumulative valid conformations of different methods during inference, with each method undergoing three independent runs. **4.** The bottom-right plot shows the $C_\alpha$-RMSD relative to the initial state along trajectories.

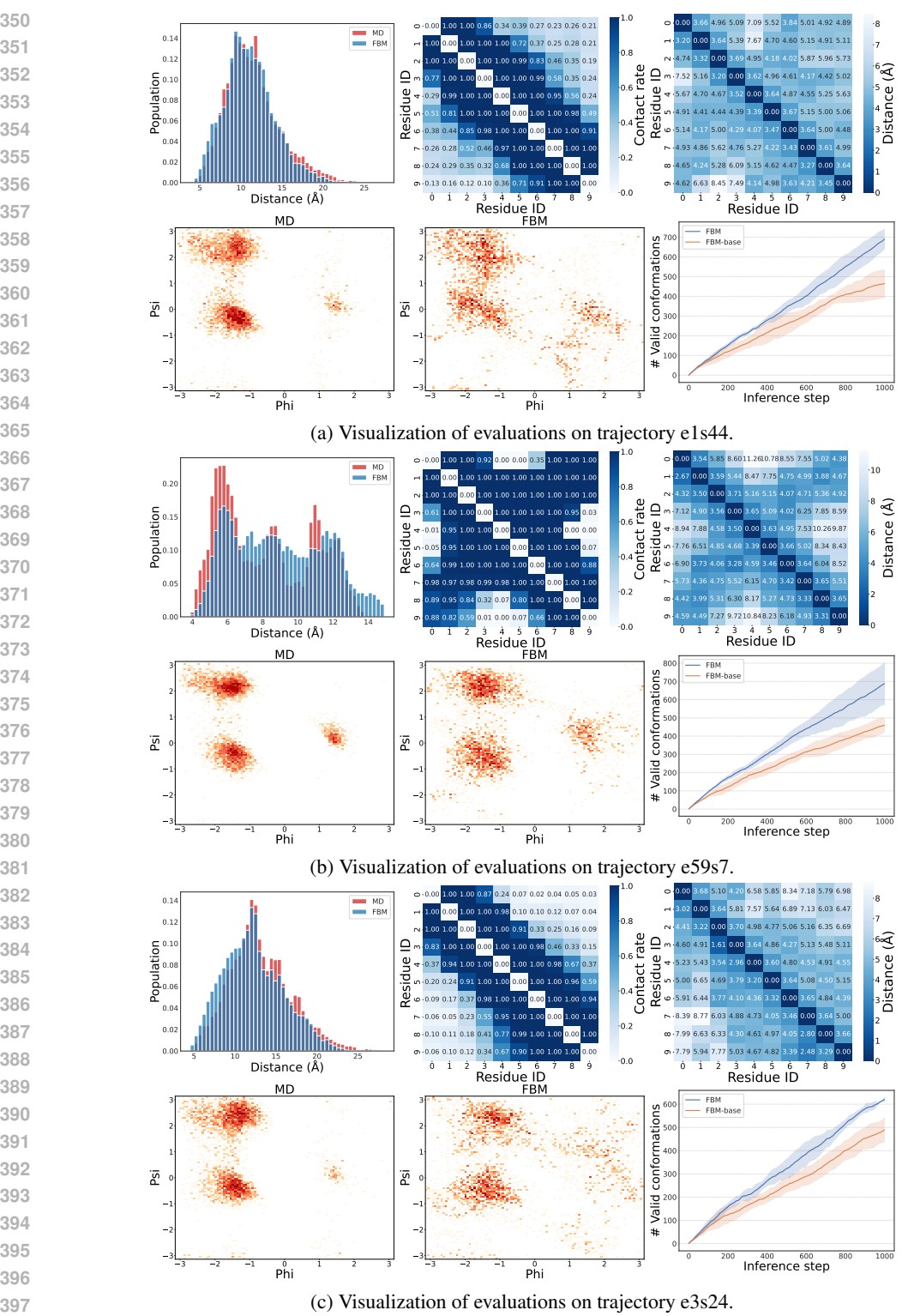

(a) Visualization of evaluations on trajectory e1s44.

(b) Visualization of evaluations on trajectory e59s7.

(c) Visualization of evaluations on trajectory e3s24.

Figure 8: The visualization of comprehensive metrics on three test trajectories of Chignolin. For each sub-figure of the corresponding peptide: **1.** The top-left plot shows the joint distribution of pairwise distances (PWD). **2.** The top-middle and top-right plots demonstrate the residue contact map and the residue minimum-distance map, respectively. **3.** The bottom-left and bottom-middle are Ramachandran plots of MD and FBM, respectively. **4.** The bottom-right plot compares the cumulative valid conformations of different methods during inference, with each method undergoing three independent runs.

