# OpenReview forum: "Force-Guided Bridge Matching for Full-Atom Time-Coarsened Dynamics of Peptides"
_ICLR.cc/2025/Conference — Submitted to ICLR 2025_

### Official Review · Reviewer_R7tW · 2024-10-28

**Soundness:** 3
**Presentation:** 3
**Contribution:** 3
**Rating:** 8
**Confidence:** 3

**Summary:**

In this paper the authors propose a model to produce relevant MD  trajectories from an initial structure, using bridge diffusion with a possibility to ensure that the recovered conformation statistic obeys a  Boltzmann distribution rather than blindly following the learned data distribution. They accomplish their goal by constraining the form of the conformation distribution to be the learned data distribution with each conformation being weighted by a learnable Boltzmann factor. So the constraints appeared from the beginning and are present in the trained model from the get-go: no need at prediction time to go through expensive re-weighting procedure, as it is part of the model. Hence their model relies on first learning the reverse and forward process that would generate the initial and final experimental conformations (conditional to those), then use this newly learned score function and use it within a dedicated network to calculate the associated Boltzmann weight (i.e. the conformation energy). At prediction time one can either just perform the simple diffusion, or decide to add the correction term to the simple diffusion-associated force field, which turns out to “simply” be the gradient of the energy term in the learned Boltzmann re-weighting coefficient: hence the name.

**Strengths:**

The paper offers a rather intuitive and straightforward way to add some physics-derived constraints to the trajectory produced at prediction time. Moreover, their model is state of the art when taking into account a rather diverse set of evaluation metrics: from comparing model output collective variables at the distribution level, in terms of validity through checking bond clashes and breaking, or finally in term structure flexibility compared to ground truth. The simplicity of adding or not the correction term in a tunable way through a guidance strength hyperparameter allows for a direct ablation study which shows the usefulness of the correction (guidance) term.
The care brought into splitting the dataset (clustering and splitting by  cluster), indeed allows them to claim generalizability, even though I  didn’t really understood why AD seems to be treated a bit separately  from the peptide in the test set.
Last but not least, it also proposes a robustness/explanatory study by adding the results of their hyperparameter tunning and some effects of those hyperparameters.

**Weaknesses:**

The paper is pretty strong and its only rather minor weakness has to do with easiness to have a direct interpretation/overall view of the metrics for the samples presented. Right now the metrics show their claims (their model is better) which is great, but they, I  believe, are hard to link to the model's performance and usefulness.
For example, I think Figure 6 is nice because it shows a direct comparison with ground truth in a more obvious way than other figures. Hence the few following questions and propositions of improvement described in the next section will be about that.

**Questions:**

Overall all those figures and metrics are complementary and I believe for a few samples to have all of those would help a lot. Indeed we have the different JS values for the full test set but having them also for the samples presented, as well as having directly the distribution comparison to the ground truth as in Figure 6 would be super helpful. I  believe it can be hard to understand what a JS value means except that it is bigger for the output of one model compared to the other.
It could be also nice to have more of those figures for your best-picked trajectories but also for those that didn’t work as well: again just to have a feel about what could be driving those numbers.
Moreover the study of hyperparameters effect (table 6) is a great add to show the robustness of the results. I believe it could be used to show the effect of force guidance on the trajectories generated and their metrics:  mainly following for a single or a few peptides how their trajectory change when the strength of guidance increases. This is touched upon in section f1 where it is pointed out that validity and contact metrics indeed nicely follow the guidance strength, but I feel like this is less true/obvious for the other metrics.
Finally, it would also be nice to see how easily this is applicable in more real case scenarios, with some scaling laws with peptide size, etc…

Overall I would say that all of my remarks are minors, but I still encourage  the authors to follow them as much as possible as I believe it will ease the overall understanding of the model ability. I would also like to state that even though I was able to follow all the proofs and didn’t see anything that I would deem to be problematic, I don’t believe I  have the level of expertise truly required to judge the math present in this paper with certainty.

---

> ### Author Response · Authors · 2024-11-21
>
> We sincerely thank your constructive comments and your recognition of our work. We address your concerns below accordingly.
>
> - **The paper is pretty strong and its only rather minor weakness has to do with easiness to have a direct interpretation/overall view of the metrics for the samples presented. ... Overall all those figures and metrics are complementary and I believe for a few samples to have all of those would help a lot.**
>
> Thank you for your valuable suggestions! In response, we have visualized comprehensive metrics, including Figures 4 and 5 in the main text and the newly added Figure 7 in the appendix. These visualizations of test peptides in different length scales intuitively demonstrate the excellent performance of FBM on OOD data and a great alignment with MD simulations.
>
> - **This is touched upon in section f1 where it is pointed out that validity and contact metrics indeed nicely follow the guidance strength, but I feel like this is less true/obvious for the other metrics.**
>
> That's a good question for discussion. In fact, to fit the true Boltzmann distribution, we should aim for an "appropriate" rather than a "larger" guidance strength. On the one hand, a larger guidance strength provides stronger physical constraints, theoretically making the model more inclined to generate reasonable conformations, which aligns with the conclusions shown in Table 6. On the other hand, an excessively large guidance strength suppresses the variability of the generative model, causing the distribution to shift toward the equilibrium Boltzmann distribution while neglecting the distribution $q_t(x_t)$ learned from time-correlated data pairs. As a result, the model may perform worse in terms of distribution similarity. This might explain the issue you raised.
>
> - **Finally, it would also be nice to see how easily this is applicable in more real case scenarios, with some scaling laws with peptide size, etc…**
>
> Thank you for your instructive suggestion and we've now conducted a tiny experiment on the smallest protein, Chignolin, and the experimental details can be found in section F.3, Table 7 and Figure 8, where FBM shows a close match to those of MD in most cases, demonstrating its usefulness and scalability to more complex molecular systems.

---

> > ### Author Response · Authors · 2024-11-24
> > **Looking forward to your reply**
> >
> > Dear Reviewer R7tW,
> >
> > We greatly appreciate the time and effort you've dedicated to reviewing our work and for your insightful questions. We hope that our responses and explanations have helped clarify the issues raised. Please let us know if you need any further information or have any additional concerns.
> >
> > Best regards.

---

> > ### Comment · Reviewer_R7tW · 2024-11-25
> >
> > Dear Authors,
> >
> > Thanks for your reply! I would like to acknowledge the huge effort that you have put into producing all those new examples with their associated better representation of their metric of interest. I believe that the picture is clearer now. So in this light I will keep my rating : good paper that should be accepted.
> > Best regards!

---

> > > ### Author Response · Authors · 2024-11-25
> > >
> > > Dear Reviewer R7tW,
> > >
> > > We sincerely appreciate your great support and insightful comments, which greatly help us make revisions to enhance the quality of our paper.
> > >
> > > Thank you once again for your valuable time and efforts!
> > >
> > > Best regards,
> > >
> > > Authors of #4521

---

### Official Review · Reviewer_oRib · 2024-11-01

**Soundness:** 3
**Presentation:** 4
**Contribution:** 4
**Rating:** 8
**Confidence:** 4

**Summary:**

The paper presents a novel model, FBM, which integrates force guidance into a bridge matching framework for time-coarsened dynamics. Traditional MD methods often struggle with achieving a balance between computational efficiency and accuracy, especially over long time scales. FBM addresses this challenge by directly targeting a Boltzmann-constrained distribution, effectively combining physical priors with generative modeling to capture full-atom time-coarsened dynamics.

**Strengths:**

- The integration of force-guided priors within the bridge matching framework for MD simulations is innovative. This approach goes beyond standard generative modeling by incorporating Boltzmann-distributed physical priors, improving the model's ability to simulate realistic molecular transitions over extended time scales.
- The authors conduct a comprehensive evaluation of FBM on two peptide datasets, demonstrating that it consistently outperforms baseline models across several metrics, including distribution similarity and validity of generated molecular conformations.
- The paper is generally well-structured, with a clear presentation of the methodology and objectives.
- The model’s ability to generate high-quality conformations with transferability across unseen systems is particularly valuable for applications requiring generalized and scalable MD simulations.

**Weaknesses:**

- The experiments to evaluate the model’s performance are conducted mainly on datasets consisting of small peptides. It would provide a more comprehensive understanding of the model's performance to extend the evaluation to include larger peptides or more complex molecular structures.
- While the FBM model demonstrates promising results, a more extensive comparison with traditional MD simulations across additional cases would strengthen its credibility. It would be helpful to provide more detailed time comparisons between FBM and MD simulations.

**Questions:**

The TIC plots in Figure 4 are not entirely clear in demonstrating the advantages of FBM. It might be more interpretable for readers to adjust the color scheme, explore alternative visualizations, or include additional quantifiable metrics.

---

> ### Author Response · Authors · 2024-11-21
>
> We sincerely thank your constructive comments and your recognition of our work. We address your concerns below accordingly.
>
> - **The experiments to evaluate the model’s performance are conducted mainly on datasets consisting of small peptides. It would provide a more comprehensive understanding of the model's performance to extend the evaluation to include larger peptides or more complex molecular structures.**
>
> Thank you for your instructive suggestion. As mentioned in the general response, we have now conducted a tiny experiment on the small protein, Chignolin. The experimental details can be found in section F.3, Table 7 and Figure 8, where FBM shows a close match to those of MD in most cases, demonstrating its usefulness and scalability to more complex molecular systems.
>
> - **While the FBM model demonstrates promising results, a more extensive comparison with traditional MD simulations across additional cases would strengthen its credibility. It would be helpful to provide more detailed time comparisons between FBM and MD simulations.**
>
> Thanks for the suggestion. We acknowledge that our initial time comparison with MD simulations was not very convincing. Therefore, we provide a more fair and reasonable comparative experiment in our revised paper. In particular, we believe that during the simulation process, both the generation speed and sample quality of the model need to be fairly considered. Therefore, we chose the effective-sample-size per second of wall-clock time as the evaluation metric for efficiency, which was also used by Timewarp [1]. The time comparison of MD simulations and FBM is shown in Figure 5.c, where FBM achieves an efficiency improvement of around 10 times relative to MD over the entire test set.
>
> - **The TIC plots in Figure 4 are not entirely clear in demonstrating the advantages of FBM. It might be more interpretable for readers to adjust the color scheme, explore alternative visualizations, or include additional quantifiable metrics.**
>
> Thanks for your advice! We've now adjusted the color scheme of TIC plots in Figure 3, 4 for higher contrast. Meanwhile, we add much more visualizations for comprehensive metrics in Figure 4, 5 and 7, including the distribution of Rg, residue contact map, cumulated valid conformations during inference, $C_{\alpha}$-RMSD over time compared to the initial states, etc. Through the visualization results of peptides of different lengths, we demonstrate that FBM exhibits excellent transferability, a strong capability to generate reasonable conformations, and is able to consistently reach the reference equilibrium distribution (1e28:C, 1n73:I, 1gxc:B), and there's still room for improvement when dealing with more complex molecular systems (1qj6:I).
>
> > [1] Klein L, Foong A, Fjelde T, et al. Timewarp: Transferable acceleration of molecular dynamics by learning time-coarsened dynamics[J]. Advances in Neural Information Processing Systems, 2024, 36.

---

> > ### Author Response · Authors · 2024-11-24
> > **Looking forward to your reply**
> >
> > Dear Reviewer oRib,
> >
> > We greatly appreciate the time and effort you've dedicated to reviewing our work and for your insightful questions. We hope that our responses and explanations have helped clarify the issues raised. Please let us know if you need any further information or have any additional concerns.
> >
> > Best regards.

---

> > > ### Comment · Reviewer_oRib · 2024-11-24
> > >
> > > Dear Authors,
> > >
> > > Thanks for your reply! I really appreciate your hard work in providing more helpful experiments and revisions. I think your reply has well solved all my questions and made the results more convincing. I will keep the current score and believe this is a good paper that should be accepted.
> > >
> > > Best,
> > > oRib

---

> > > > ### Author Response · Authors · 2024-11-25
> > > >
> > > > Dear Reviewer oRib,
> > > >
> > > > We sincerely appreciate your great support and insightful comments, which greatly help us make revisions to enhance the quality of our paper.
> > > >
> > > > Thank you once again for your valuable time and efforts!
> > > >
> > > > Best regards,
> > > >
> > > > Authors of #4521

---

### Official Review · Reviewer_8Atg · 2024-11-01

**Soundness:** 2
**Presentation:** 3
**Contribution:** 1
**Rating:** 5
**Confidence:** 5

**Summary:**

The paper proposes a conditional generative model called Force-guided Bridge Matching (FBM) to improve Molecular Dynamics (MD) simulations. Traditional MD simulations, while accurate, are time-consuming due to the need for small time steps. FBM leverages a physics-informed approach to time-coarsened dynamics by incorporating an intermediate force field to guide the generation process. This allows the model to better approximate the Boltzmann distribution, essential for realistic molecular conformations, and improves simulation efficiency without sacrificing accuracy.

**Strengths:**

1. The paper is clearly written and easy to follow, also this task is indeed important.
2. The experiments are effectively support the conclusions.
3. The code and detailed hyperparameters are provided, making it easy for others to reproduce the results.

**Weaknesses:**

I believe the main weaknesses with this paper is its lack of novelty, as it appears to be a straightforward combination of [1], [2] and [3] in a slightly altered setting. It seems the author did not conduct a comprehensive literature review. My reasons are as follows:

1. Training with trajectory pairs was already introduced in [1]; the only difference here is that [1] focused on protein backbones and employed SE(3) FM. In addition, next frame prediction is also a subtask in [2].
2. The concept of force-guided training was discussed in [3], with the main difference being that [3] uses SE(3) diffusion.

In essence, this paper combines aspects of both [1] ([2]) and [3] without properly referencing [1], [2]. While it presents a solid engineering application, it lacks originality.

Additionally, I have another concerns, which I’ve outlined in the Questions section.

>[1] Li, S., Wang, Y., Li, M., Shao, B., Zheng, N., Jian, Z., & Tang, J. F $^ 3$ low: Frame-to-Frame Coarse-grained Molecular Dynamics with SE (3) Guided Flow Matching. In ICLR 2024 Workshop on Generative and Experimental Perspectives for Biomolecular Design.

>[2] Jing, B., Stark, H., Jaakkola, T., & Berger, B. Generative Modeling of Molecular Dynamics Trajectories. In ICML'24 Workshop ML for Life and Material Science: From Theory to Industry Applications.

>[3] Wang, L., Shen, Y., Wang, Y., Yuan, H., Wu, Y., & Gu, Q. Protein Conformation Generation via Force-Guided SE (3) Diffusion Models. In Forty-first International Conference on Machine Learning.

**Questions:**

1. Using bridge matching may not be ideal, as all samples along the same trajectory are expected to follow the same Boltzmann distribution. Ideally, we would achieve **a constant transformation**; however, in practice, we are performing point-to-point correspondence (from one delta distribution to another). This approach, therefore, appears theoretically unsound. What are your thoughts on this?
2. Experiments on peptides alone are too limited and should at least be performed on the smallest “protein” (backbone), Chignolin.

---

> ### Author Response · Authors · 2024-11-21
>
> We sincerely thank the reviewer for the comments. We respectfully disagree that our paper lacks novelty and is a straightforward combination of [1], [2] and [3]. The reviewer probably misunderstood our claims and contributions. We provide the reasons below.
>
> - **Training with trajectory pairs was already introduced in [1]; the only difference here is that [1] focused on protein backbones and employed SE(3) FM. In addition, next frame prediction is also a subtask in [2].**
>
> Thank you for bringing these two references [1,2] to our attention. It seems there might be a misunderstanding regarding our claims. We do not assert that we are the first to perform MD simulations using "training with trajectory pairs" or "next-frame prediction". In fact, in the third paragraph of the Introduction, we explicitly acknowledge that prior works, such as Timewarp [4], have already employed this training strategy, albeit under a different terminology, namely "time-coarsened dynamics simulation." Our intended claim is that our method enhances these existing approaches by incorporating the underlying Boltzmann constraint and physics priors, which are not addressed in [1,2].
>
> We also apologize for initially overlooking the two works mentioned [1,2], as they are both workshop papers. In response to the reviewer's feedback, we have now properly cited and discussed these works in the Introduction and Related Work sections of our revised manuscript. Since both methods take a coarse-grained approach (F$^3$low [1] focuses on protein backbones, and MDGen [2] treats backbone atoms as rigid bodies), we did not implement them in our experiments to ensure fair comparisons.

---

> ### Author Response · Authors · 2024-11-21
>
> - **The concept of force-guided training was discussed in [3], with the main difference being that [3] uses SE(3) diffusion.**
>
> We appreciate the reviewer’s observation that the concept of force-guided training draws inspiration from [3]. This connection was honestly and explicitly acknowledged in our Related Work section. However, we would like to emphasize that our adoption of the bridge-matching model as a replacement for the diffusion model is both well-motivated and nontrivial.
>
> From a problem-solving perspective, the bridge-matching model is better suited to our MD task. While the approach in [3], which employs a diffusion model, is effective for generating conformation ensembles, it does not account for the temporal correlation between them. Consequently, it cannot be utilized to study dynamic processes. For example, proteins may undergo folding and unfolding transitions under varying environmental conditions [4]. While conformation ensembles can capture different states, they cannot characterize the transitions between these states. To address this limitation and effectively learn transition probabilities between temporally correlated data pairs, we chose bridge matching. This approach generates molecular conformations conditioned on the starting conformations rather than relying on Gaussian priors, making it a more appropriate choice for the MD task.
>
> From a methodological perspective, conducting force-guided training for the bridge-matching model is far from trivial. A core aspect of deep generative models is the learning of the generation process. For the diffusion model used in [3], this process (the backward process) is driven by the score function $\nabla\log{p_t(x_t)}$, which can be straightforwardly derived using Eq. (6) in [3]. Additionally, the immediate force field $\nabla_{x_t}\varepsilon_t(x_t)$ involved in the computation is directly obtainable via Eq. (5) in [3]. In contrast, in our work, the generation process for the bridge matching model is governed by the vector field $v'(x_t, t)$, which is significantly more challenging to derive. To address this, we should not only establish the relationship between the vector field, the score function and the immediate force field through Propositions 3.1, 3.2 and 3.3, but also rely on the assumptions of symmetric Brownian bridge and the detailed balance of the stationary distribution of MD simulations. While the final form appears relatively simple, the derivation process is intricate, as detailed in the proof provided in Section B of our paper.
>
> Furthermore, the immediate force field involved in Proposition 3.1 has a more complex form compared to that of [3] (please compare Eq. (9) in our paper with Eq. (5) in [3]). This is basically because both ends of bridge matching are data distributions, while in diffusion only one end is a data distribution and another is Gaussian. Computing the force field necessitates the score $\nabla\log{q_t(x_t)}$, which, in turn, requires introducing the vector field of the reverse-time Brownian bridge (Eqs. (11–12)) and the use of Proposition 3.3.
>
> As such, our force-guided bridge-matching approach is not a simple substitution for the diffusion model. Instead, it embodies theoretical insights and involves nontrivial derivations, making it a meaningful and innovative contribution to the field.
>
> - **In essence, this paper combines aspects of both [1] ([2]) and [3] without properly referencing [1], [2]. While it presents a solid engineering application, it lacks originality.**
>
> Overall, we acknowledge that our task (i.e., "forward simulation" or "time-coarsed dynamics") is consistent with that of [1] and [2], yet we did not claim that the task was first introduced by us.
> Similarly, we recognize that the concept of force-guided training is inspired by [3]. That said, applying force-guided training to our task poses unique challenges. We opted for the bridge-matching model over the diffusion model as it is better suited for MD tasks. To integrate the force field into the bridge matching framework, we have developed several insightful propositions and advanced techniques, which we believe represent meaningful theoretical and practical contributions.

---

> ### Author Response · Authors · 2024-11-21
>
> - **Using bridge matching may not be ideal, as all samples along the same trajectory are expected to follow the same Boltzmann distribution. Ideally, we would achieve a constant transformation; however, in practice, we are performing point-to-point correspondence (from one delta distribution to another). This approach, therefore, appears theoretically unsound. What are your thoughts on this?**
>
> Thanks for your concern and it deserves further discussion. We believe that the statements "the distributions at both ends of the bridge are ideally Boltzmann distributions" and "bridge matching performs point-to-point correspondence" are not contradictory.
>
> Firstly, given a temporal interval $\tau$, we select multiple data pairs $(x_0,x_1)$ from the MD trajectories, which are temporally correlated and thus not independent. However, this does not affect the fact that the marginal distributions of their joint distribution are ideally Boltzmann distributions, i.e., $\int{q(x_0, x_1)dx_0}=\int{q(x_0,x_1)dx_1=\exp(-kU(x))}$.
>
> Secondly, as a Markov process, bridge matching actually learns the expectation of the posterior distribution $q_t(x_0,x_1|x_t)$ given the state $x_t$ at diffusion time $t$ from the joint data distribution $q(x_0, x_1)$. The equation converges to a delta distribution as $t\to{0}$ or $t\to{1}$ only when the bridge is pinned down to two given starting and ending states (i.e., during the training process). Therefore, in the ideal case, the distribution learned by bridge matching at $t=0$ (or $t=1$) is the expectation of the delta distribution, which is "luckily" identical to the marginal distributions $q(x_0)$ and $q(x_1)$. Taking all the above into consideration, bridge matching is compatible with the assumption of the stationary distribution in MD simulations.
>
> - **Experiments on peptides alone are too limited and should at least be performed on the smallest “protein” (backbone), Chignolin.**
>
> Thanks for your suggestion! We have now conducted a tiny experiment on Chignolin, and the experimental details can be found in section F.3, Table 7 and Figure 8, where FBM shows a close match to those of MD in most cases, demonstrating its usefulness and scalability to more complex molecular systems.
>
> > [1] Li, S., Wang, Y., Li, M., Shao, B., Zheng, N., Jian, Z., & Tang, J. F 3 low: Frame-to-Frame Coarse-grained Molecular Dynamics with SE (3) Guided Flow Matching. In ICLR 2024 Workshop on Generative and Experimental Perspectives for Biomolecular Design.
>
> > [2] Jing, B., Stark, H., Jaakkola, T., & Berger, B. Generative Modeling of Molecular Dynamics Trajectories. In ICML'24 Workshop ML for Life and Material Science: From Theory to Industry Applications.
>
> > [3] Wang, L., Shen, Y., Wang, Y., Yuan, H., Wu, Y., & Gu, Q. Protein Conformation Generation via Force-Guided SE (3) Diffusion Models. In Forty-first International Conference on Machine Learning.
>
> > [4] Klein L, Foong A, Fjelde T, et al. Timewarp: Transferable acceleration of molecular dynamics by learning time-coarsened dynamics[J]. Advances in Neural Information Processing Systems, 2024, 36.
>
> > [5] Lindorff-Larsen K, Piana S, Dror R O, et al. How fast-folding proteins fold[J]. Science, 2011, 334(6055): 517-520.
>
> > [6] Anderson B D O. Reverse-time diffusion equation models[J]. Stochastic Processes and their Applications, 1982, 12(3): 313-326.

---

> > ### Author Response · Authors · 2024-11-24
> > **Looking forward to your reply**
> >
> > Dear Reviewer 8Atg,
> >
> > We greatly appreciate the time and effort you've dedicated to reviewing our work and for your insightful questions. We hope that our responses and explanations have helped clarify the issues raised. Please let us know if you need any further information or have any additional concerns.
> >
> > Best regards.

---

> > > ### Comment · Reviewer_8Atg · 2024-11-25
> > >
> > > Thank you for your rebuttal. I would like to point out that flow matching and diffusion models are, to some extent, unified (meaning the score function and vector field are somewhat equivalent). Therefore, I still believe the force-guided idea lacks sufficient novelty.
> > >
> > > Additionally, in your rebuttal, the bridge matching works because $q(x_0)$ and $q(x_1)$ are treated as distinct delta distributions. However, my question assumes that $q(x_0)$ and $q(x_1)$ follow Boltzmann distributions. In this case, introducing bridge matching does not seem ideal. It might be more convincing to condition on $x_0$ and sample $x_1$ directly from the Boltzmann distribution.
> > >
> > > That said, I appreciate the chignolin experiments, which strengthen the paper overall. Based on this, I have decided to raise my score to 5.

---

> ### Author Response · Authors · 2024-11-25
>
> Dear reviewer 8Atg,
>
> We sincerely appreciate your great support and insightful comments, and we are also very grateful for your increased evaluation of our work. Here we would like to address your additional concerns, which may help facilitate a better understanding and a more in-depth discussion.
>
> - **I would like to point out that flow matching and diffusion models are, to some extent, unified (meaning the score function and vector field are somewhat equivalent).**
>
> We fully agree with your statement that the vector field and score are equivalent to some extent, which have been proven by Proposition 3.3. However, the main difference arises from the fact that diffusion has a simple Gaussian prior at one end, while both ends of bridge matching are data distributions. This leads to the diffusion SDE process being controlled by **a single variable** (e.g., the score function), while bridge matching must be described by two variables (e.g., $v_{\mathrm{fwd}}$ and $v_{\mathrm{rev}}$ in our work). To be more specific, both the reverse-time diffusion and bridge matching can be unified with the following SDE process, where we use the notations in [1]:
>
> $dx_t=b(t,x_t)dt+\epsilon(t)s(t,x_t)dt+\sqrt{2\epsilon(t)}dW_t,$
>
> where $s(t,x_t)$ is the score function at diffusion time $t$, $b(t,x_t)=-\frac{1}{2}\beta_tx_t$ as in [2] while $b(t,x_t)=\frac{v_{\mathrm{fwd}}(t,x_t)+v_{\mathrm{rev}}(t,x_t)}{2}$ in our bridge matching framework. It is obvious that, though the score function $s(t,x_t)$ is considered in both methods, the drift term $b(t,x_t)$ has a quite non-trivial form in bridge matching. In fact, most of our effort has been dedicated to proving that, after incorporating the Boltzmann constraint, $b'(t,x_t)=b(t,x_t)$ holds (Proposition 3.2), **which is not considered at all in diffusion**. We believe this is the main challenge and contribution of incorporating the Boltzmann constraint into bridge matching.
>
> - **Additionally, in your rebuttal, the bridge matching works because $q(x_0)$ and $q(x_1)$ are treated as distinct delta distributions. However, my question assumes that $q(x_0)$ and $q(x_1)$ follow Boltzmann distributions. In this case, introducing bridge matching does not seem ideal. It might be more convincing to condition on $x_0$ and sample $x_1$ directly from the Boltzmann distribution.**
>
> Our rebuttal may have caused some misunderstandings and we would like to clarify it here. Our statement is, **given the bridge pinned down to two endpoints $x_0,x_1$, i.e., in the training process**, the learned forward vector field $v_{\mathrm{fwd}}$, which is equivalent to $q_t(x_1|x_t)$, will converge to $q(x_1|x_t)\delta(x_t-x_0)$ when the diffusion time $t$ converges to 0. Therefore, in the ideal case, the model learns **the expectation of the delta distribution** when $t=0$ after the whole training process. That is, $\mathbb{E}_{x_0}q(x_1|x_t)\delta(x_t-x_0)=q(x_1|x_0)$, where $q(x_1|x_0)$ is the transition probability estimated on the training dataset. Thus, if we assume "$q(x_0)$ and $q(x_1)$ follow Boltzmann distributions" in the ideal case, our model will learn the transition probability of the Boltzmann distribution accordingly. Thus we believe that the assumption of Boltzmann distributions and the bridge matching framework are indeed compatible.
>
> By the way, we also believe that, in the case of randomly selecting data pairs from the original MD trajectories for training, the assumption of both ends following the Boltzmann distribution is unlikely to hold. In such non-ideal circumstances, introducing the Boltzmann constraint helps bring the generated distribution closer to the Boltzmann distribution, which is also the reason we incorporate the force guidance into the bridge matching framework.
>
> We hope that our response helps to further address your concerns.
>
> Best regards,
>
> Authors of #4521
>
> Reference
>
> > [1] Albergo M S, Boffi N M, Vanden-Eijnden E. Stochastic interpolants: A unifying framework for flows and diffusions[J]. arXiv preprint arXiv:2303.08797, 2023.
>
> > [2] Wang Y, Wang L, Shen Y, et al. Protein conformation generation via force-guided se (3) diffusion models[J]. arXiv preprint arXiv:2403.14088, 2024.

---

> > ### Author Response · Authors · 2024-11-29
> >
> > Dear Reviewer 8Atg,
> >
> > We thank for your insightful questions about our bridge matching framework, and we manage to provide accurate explanations accordingly. Please let us know if you have other concerns that need further discussion.
> >
> > Best regards,
> >
> > Authors of #4521

---

### Author Response · Authors · 2024-11-21
**General Response**

We thank all reviewers for their valuable comments. We have carefully revised our paper to address some common concerns listed below. Note that all the revisions in our updated paper have been highlighted in red.

- Responding to the concerns raised by Reviewer 8Atg about the originality of our work, we explain that this work not only provides solid engineering results but also presents a theoretical contribution by first integrating an intermediate force field based on the Boltzmann constraint into the bridge matching framework. We outline the necessity of using bridge matching to model the temporal correlations in dynamic processes, explain how bridge matching is compatible with the equilibrium distribution of MD, and discuss that, despite the simplicity of the FBM generation process, it is a non-trivial derivation that effectively combines multiple prior assumptions.
- Responding to the suggestions from Reviewer 8Atg, oRib, R7tW that experiments should be extended to include more complex molecular structures, we conduct a tiny experiment for the smallest protein, Chignolin. We fine-tune our model to the MD dataset of Chignolin curated by [1], then we evaluate our FBM on three other independent trajectories with different initial states. The details of experimental settings and results are reported in Section F.3, Table 7 and Figure 8 in our revised manuscripts. Here, we briefly present the results on three test trajectories, where FBM significantly outperforms FBM-base across multiple comprehensive metrics, showcasing a strong and stable generation ability for molecular dynamics。

| Index | Model    |   P{\small W}D  |   R{\small G}   |       TIC       |   R{\small AM}  | V{\small AL}-CA ($\uparrow$) | C{\small ONTACT} ($\downarrow$) |
|-------|----------|:---------------:|:---------------:|:---------------:|:---------------:|:----------------------------:|:-------------------------------:|
| e1s44 | FBM      | **0.315**/0.056 | **0.285**/0.080 | **0.544**/0.011 |   0.509/0.029   |        **0.691**/0.041       |         **0.161**/0.069         |
|       | FBM-base |   0.417/0.069   |   0.467/0.148   |   0.554/0.023   | **0.480**/0.013 |          0.465/0.057         |           0.249/0.066           |
| e59s7 | FBM      | **0.395**/0.017 | **0.400**/0.034 | **0.522**/0.015 | **0.443**/0.020 |        **0.780**/0.012       |         **0.184**/0.029         |
|       | FBM-base |   0.456/0.026   |   0.407/0.016   |   0.526/0.002   |   0.490/0.017   |          0.460/0.031         |           0.219/0.056           |
| e3s24 | FBM      | **0.305**/0.046 | **0.332**/0.089 | **0.524**/0.004 |   0.519/0.011   |        **0.628**/0.010       |         **0.123**/0.032         |
|       | FBM-base |   0.450/0.086   |   0.549/0.149   |   0.527/0.015   | **0.502**/0.019 |          0.490/0.040         |           0.278/0.104           |

- Reviewer oRib suggests that it would be helpful to provide more detailed time comparisons between FBM and MD simulations. We believe that during the simulation process, both the generation speed and sample quality of the model need to be fairly considered. Therefore, we chose the effective-sample-size per second of wall-clock time as the evaluation metric for efficiency, which was used by Timewarp [2]. The comparison of MD simulations and FBM is shown in Figure 5.c, where FBM achieves an efficiency improvement of around 10 times relative to MD over the entire test set.
- Reviewer oRib suggests that the color scheme should be adjusted for better comparison, and Reviewer oRib, R7tW believe more visualizations of comprehensive metrics would be super helpful. In our revised paper, we have adjusted the color scheme of TIC plots of Figure 3 and 4 for higher contrast. We also made significant revisions to Figures 4 and 5 that provides visualizations of comprehensive metrics, including: the distribution of the radius-of-gyration (Rg), the residue contact map, cumulated valid conformations during inference, $C_{\alpha}$-RMSD values along the trajectory compared to initial states, and the effective-sample-size per second compared to MD simulations. Meanwhile, the visualization of 3 other test peptides was illustrated in Figure 7, Section F.2, where we show FBM demonstrates super transferability to unseen peptides with different residue lengths.

We would give further responses to other specific concerns to each reviewer individually.

> [1] Pérez Culubret, Adrià; De Fabritiis, Gianni (2021). Chignolin Simulations. figshare. Dataset. https://doi.org/10.6084/m9.figshare.13858898.v1.

> [2] Klein L, Foong A, Fjelde T, et al. Timewarp: Transferable acceleration of molecular dynamics by learning time-coarsened dynamics[J]. Advances in Neural Information Processing Systems, 2024, 36.

---

### Comment · Area_Chair_6FCs · 2024-11-25

Dear reviewers,

Tomorrow (Nov 26) is the last day for asking questions to the authors. With this in mind, If you have not already done so, please read the latest update to the rebuttal provided by the authors earlier today, as well as the other reviews. If you still have outstanding comments or questions, please provide your updated view _accompanied by a motivation_ and raise any outstanding questions for the authors.

**Timeline**: As a reminder, the review timeline is as follows:
- November 26: Last day for reviewers to ask questions to authors.
- November 27: Last day for authors to respond to reviewers.
- November 28 - December 10: Reviewer and area chair discussion phase.

Thank you for your hard work,

Your AC

---

### Meta-Review · Area_Chair_6FCs · 2024-12-19

**Metareview:**

This paper received mixed scores, with two reviewers leaning to a clear accept (score 8), and one reviewer initially learning towards recommending to reject this paper (score 3). Among other things, the paper was thought to have a clear presentation, and claims that were validated with their empirical results, which included evaluations on alanine dipeptide and small dipeptides. As areas for improvement, points were raised on the need for validation on more than just small peptides and the need for timing comparisons. Furthermore, the reviewer who recommended rejection indicated that they viewed the work as a straightforward combination of existing concepts in prior work, and that the work did not show an advantage through this combination.

During the rebuttal, the authors have added additional experiments for the small protein Chignolin, where they compare their model against MD only. Additionally, the authors included comparisons of effective sample size per second of wall-clock time as an additional metric for efficiency, which was introduced in prior work.

Based on this rebuttal and following discussions the two reviewers in favor of acceptance maintained their score, while the reviewer with the most concerns raised their score from a 3 to a 5, based on the remaining concern of limited novelty.

After the author-reviewer discussion ended, I read through the reviews again, as well as the discussions with the authors and the newly computed results and the revised paper. Upon going through the revised paper to look through the added results, I found that the results on alanine dipeptide raise questions about their correctness. In particular, the distribution of samples from the models in figure 3 do not seem to resemble the distribution represented by the contours based on kernel density estimation of MD trajectories, and the baseline methods in particular do not perform well at all. This discrepancy between model distributions and the MD-based distribution is striking since one of the baseline methods, Timewarp, has shown in their original paper (https://arxiv.org/pdf/2302.01170) to be able to draw samples of alanine dipeptide according to a distribution that closely mimics the distribution of samples from MD trajectories. See figure 3 in the Timewarp paper. This raises doubts for me on whether or not the results in this current submission were calculated correctly.
After consulting with the reviewers, the reviewers agreed that the results raise concerns about correctness. Unfortunately, the range of angles used to compute Ramachandran plots of alanine dipeptide in this submission were not displayed, nor are samples from the reference MD simulation included in the Ramachandran plots. This makes it hard to determine if the discrepancy between the results reported in this paper compared to the Timewarp paper are influenced by different data or plotting settings. Moreover, we received input from one additional reviewer with significant expertise in this field, who was initially unable to review the paper but looked through the results and other reviews. In addition to sharing concerns about the alanine dipeptide results, they noted that the TICA plots in figure 4 show worryingly bad results, and that the energies in Figure 6 are also not good. Unfortunately, during the reviewer-AC discussion phase it was no longer possible to ask for clarification from the authors, and the presented results did not have enough details to be able to determine where the discrepancies with previously reported results come from. Given the overall consensus among reviewers and AC that the presented results raise concerns, I recommend rejecting this paper.

I would like to thank the authors for submitting the paper and for providing additional results during the rebuttal. I hope the above explanation can be helpful in rewriting the manuscript for a future submission.

**Additional Comments On Reviewer Discussion:**

See above.

---

### Decision · Program_Chairs · 2025-01-22

Reject